# Activation of the ubiquitin-proteasome system contributes to oculopharyngeal muscular dystrophy through muscle atrophy

**Cécile Ribot**[1☯], **Cédric Soler**[1☯¤], **Aymeric Chartier**[1☯], **Sandy Al Hayek**[2], **Rima Naït-Saïdi**[1], **Nicolas Barbezier**[1], **Olivier Coux**[3], **Martine Simonelig**[1]*

**1** mRNA Regulation and Development, Institute of Human Genetics, UMR9002 CNRS-Univ Montpellier, Montpellier, France, **2** GReD Laboratory, Clermont-Auvergne University, INSERM U1103, CNRS UMR6293, Clermont-Ferrand, France, **3** Ubiquitin-proteasome system and cell cycle control, Montpellier Cell Biology Research Center, UMR5237 CNRS-Univ Montpellier, Montpellier, France

☯ These authors contributed equally to this work.
¤ Current address: GReD Laboratory, Clermont-Auvergne University, INSERM U1103, CNRS UMR6293, Clermont-Ferrand, France
* Martine.Simonelig@igh.cnrs.fr

**Data Availability Statement:** All relevant data are within the manuscript and its Supporting Information files.

## Abstract

Oculopharyngeal muscular dystrophy (OPMD) is a late-onset disorder characterized by progressive weakness and degeneration of specific muscles. OPMD is due to extension of a polyalanine tract in poly(A) binding protein nuclear 1 (PABPN1). Aggregation of the mutant protein in muscle nuclei is a hallmark of the disease. Previous transcriptomic analyses revealed the consistent deregulation of the ubiquitin-proteasome system (UPS) in OPMD animal models and patients, suggesting a role of this deregulation in OPMD pathogenesis. Subsequent studies proposed that UPS contribution to OPMD involved PABPN1 aggregation. Here, we use a *Drosophila* model of OPMD to address the functional importance of UPS deregulation in OPMD. Through genome-wide and targeted genetic screens we identify a large number of UPS components that are involved in OPMD. Half dosage of UPS genes reduces OPMD muscle defects suggesting a pathological increase of UPS activity in the disease. Quantification of proteasome activity confirms stronger activity in OPMD muscles, associated with degradation of myofibrillar proteins. Importantly, improvement of muscle structure and function in the presence of UPS mutants does not correlate with the levels of PABPN1 aggregation, but is linked to decreased degradation of muscle proteins. Oral treatment with the proteasome inhibitor MG132 is beneficial to the OPMD *Drosophila* model, improving muscle function although PABPN1 aggregation is enhanced. This functional study reveals the importance of increased UPS activity that underlies muscle atrophy in OPMD. It also provides a proof-of-concept that inhibitors of proteasome activity might be an attractive pharmacological approach for OPMD.

**Funding:** This work was supported by the CNRS-University of Montpellier UMR9002, AFM-Téléthon eOPMD project 17110, Fondation pour la Recherche Médicale ("Equipe FRM 2013 DEQ20130326534"), Agence Nationale pour la Recherche (ANR-09-GENO-025-01 from GIS-Maladies Rares and AFM-Téléthon, and ANR-17-CE12-0011-01) to MS. CR, CS and NB were supported by the ANR-GIS-Maladies Rares (ANR-09-GENO-025-01); SAH was supported by ISITE CAP2025 Challenge 3; RNS was supported by AFM-Téléthon (17110). The funders had no role in study design, data collection and analysis, decision to publish, or preparation of the manuscript.

**Competing interests:** I have read the journal's policy and the authors of this manuscript have the following competing interests: The authors declare that AC (2%) and MS (3%) are co-inventors of the patent "Proteasome inhibitors for treating a disorder related to an accumulation of a nondegraded abnormal protein or a cancer", WO/2016/113357 that has been published on July 21, 2016.

## Author summary

Oculopharyngeal muscular dystrophy (OPMD) is a genetic disease characterized by progressive weakness of specific muscles, leading to swallowing difficulties (dysphagia), eyelid drooping (ptosis) and walking difficulties at later stages. No drug treatments are currently available. OPMD is due to mutations in a nuclear protein called poly(A) binding protein nuclear 1 (PABPN1) that is involved in processing of different classes of RNAs in the nucleus. We have used an animal model of OPMD that we have developed in the fly *Drosophila* to investigate the role in OPMD of the ubiquitin-proteasome system, a pathway specialized in protein degradation. We report an increased activity of the ubiquitin-proteasome system that is associated with degradation of muscular proteins in the OPMD *Drosophila* model. We propose that higher activity of the ubiquitin-proteasome system leads to muscle atrophy in OPMD. Importantly, oral treatment of this OPMD animal model with an inhibitor of proteasome activity reduces muscle defects. A number of proteasome inhibitors are approved drugs used in clinic against cancers, therefore our results provide a proof-of-concept that inhibitors of proteasome might be of interest in future treatments of OPMD.

## Introduction

Protein aggregation represents a pathological hallmark in many neurodegenerative diseases, including Alzheimer disease, Parkinson disease, Huntington disease and amyotrophic lateral sclerosis, among others. A key feature of all these diseases is the accumulation of abnormally processed and misfolded proteins that form aggregates and partly lose their physiological roles [1].

Oculopharyngeal muscular dystrophy (OPMD) is one of these proteinopathies in which the mutant protein accumulates as aggregates in diseased nuclei. OPMD is a rare autosomal dominant muscular dystrophy that starts in the late fifties. It is characterized by progressive weakness of specific muscles, leading to eyelid drooping (ptosis), swallowing difficulties (dysphagia), and proximal limb weakness [2,3]. OPMD is due to short expansions of a GCN repeat in the gene encoding poly(A) binding protein nuclear 1 (PABPN1) [4]. Triplet expansion in *PABPN1* leads to extension of a polyalanine tract at the N-terminus of the protein from 10 alanines in the normal protein, to 11 to 18 alanines in patients [5,6]. Alanine-expanded PABPN1 form nuclear aggregates in muscle fibres, which are a pathological hallmark of the disease [7].

PABPN1 is a nuclear protein at steady-state and has several molecular functions in RNA processing and surveillance. First identified for its role in nuclear polyadenylation, a reaction in which PABPN1 stimulates poly(A) polymerase and controls poly(A) tail length [8–12], it is also involved in alternative polyadenylation by binding to proximal weak poly(A) sites and preventing their utilization [13]. More recently, PABPN1 was found to be involved in nuclear RNA decay through the recruitment of the exosome or the PARN deadenylase [14,15]. These decay functions of PABPN1 include polyadenylation-dependent processing and turnover of long non-coding RNAs and small nucleolar RNAs [16,17], nuclear surveillance through hyperadenylation and decay of RNAs retained in the nucleus [18], and 3'-end maturation of the human telomerase RNA [19–21].

Defects in one or several of these functions are expected to be the initial trigger of OPMD. Consistent with this, a general shift towards proximal poly(A) site utilization and upregulation of shorter mRNAs produced through this poly(A) site shift have been described in a mouse

model of OPMD [13,22]. More recently, using *Drosophila* and mouse models of OPMD, we showed that a primary defect in OPMD is in nuclear cleavage and polyadenylation. This defect is general and leads in turn to specific poly(A) tail shortening of mRNAs encoding mitochondrial proteins, because this class of mRNAs are more actively deadenylated by the CCR4-NOT deadenylation complex [23]. This molecular defect leads to reduced levels of mitochondrial proteins in OPMD patient muscles, resulting in decreased mitochondrial activity and is among the earliest defects in OPMD pathogenesis, occurring in presymptomatic patient muscles [23]. Decreased levels of mitochondrial proteins as well as poly(A) tail shortening was also found in another OPMD mouse model that closely reproduces *Pabpn1* mutation in OPMD patients, i.e. *Pabpn1-17ala/Pabpn1* at the endogenous *Pabpn1* locus [24].

Polyalanine expansions in PABPN1 induces its aggregation. However, although PABPN1 aggregates are a key feature of OPMD, how these aggregates contribute to the disease remains poorly understood. mRNAs, as well as several RNA binding proteins, including hnRNP proteins and poly(A) polymerase, are recruited to PABPN1 aggregates [25–28]. In addition, more recent studies have shown that wild-type PABPN1 is also sequestered in aggregates [13], thus resulting in reduced levels of soluble protein that might contribute to the pathology through a loss of function mechanism [24,29]. In favor of this model, some of the molecular defects in OPMD are consistent with a loss of function of PABPN1 [13,23]. In addition, reducing PABPN1 aggregation load through either oral treatment with anti-aggregation drugs, or expression of anti-PABPN1 intrabody improves muscle function in *Drosophila* and mouse OPMD models [30–34].

Importantly, HSP70 and components of ubiquitin-proteasome system (UPS), i.e. ubiquitin and subunits of the proteasome, are also recruited to PABPN1 nuclear aggregates [25], suggesting that mutant forms of PABPN1 are targeted by the ubiquitin-proteasome degradation pathway. In lines with this, the stability of both wild-type and alanine-expanded PABPN1 in cultured cells was shown to depend on the UPS, and an E3 ubiquitin ligase specific to PABPN1, ARIH2 has been identified [35,36].

In another study, an integrated RNA profiling analysis including *Drosophila* and mouse models of OPMD as well as patient samples showed that the UPS was the most consistently and significantly deregulated pathway in OPMD [37]. These data suggested the contribution of UPS-dependent protein homeostasis in OPMD pathogenesis. UPS function in protein homeostasis relies on an enzymatic cascade of ubiquitination and degradation steps. Ubiquitination requires ubiquitin that is first activated by the ubiquitin-activating enzyme E1 and then transferred to an E2-conjugating enzyme. Ubiquitin is subsequently conjugated to target proteins through the action of E3 ligases that are responsible for target specificity. Processing of poly-ubiquitinated (poly-Ub) proteins involves deubiquitinating enzymes (DUBs) that can remove ubiquitin from them and the proteasome that degrade the poly-Ub targets into small peptides [38]. Consistent deregulation in OPMD animal models and patient samples was found for E3 ligases, DUBs and proteasome subunits, with E3 ligases being mostly upregulated [37].

In neurodegenerative proteinopathies, insufficient clearance of misfolded proteins through protein quality control pathways such as autophagy and the UPS, is recognized to be a major cause of pathogenesis [1]. Increasing protein clearance by enhancing activity of protein degradation pathways, either with small molecules or genetically, can reduce neurodegeneration in animal models [1,39].

In contrast, here we show, using a *Drosophila* model of OPMD, that muscle weakness and degeneration in OPMD depends on increased proteasomal activity. Specific expression of alanine-expanded mammalian PABPN1 in *Drosophila* muscles reproduces the main features of OPMD: progressive muscle weakness and degeneration resulting in *Drosophila* wing position

defects, and accumulation of mutant PABPN1 in nuclear aggregates in affected muscles [31,40]. Here, we have used an OPMD *Drosophila* model induced by mesodermal expression of alanine-expanded PABPN1 to perform a genome-wide genetic screen with large deletions that cover most of the *Drosophila* genome. This screen and a secondary screen focused on UPS components identified a large number of UPS mutants as suppressors of OPMD defects. We showed that both proteasome activity and levels of ubiquitinated myofibrillar proteins are increased in OPMD *Drosophila* muscles. Accordingly, myofibrillar protein levels were reduced in OPMD muscles. Finally, reducing proteasomal activity using oral pharmacological treatment reduced OPMD wing position defects, although the PABPN1 aggregation load was increased. We propose that deregulation of the UPS has a major contribution to OPMD pathogenesis. This deregulation leads to increased proteasomal activity and degradation of myofibrillar proteins. Importantly, the role of the UPS in OPMD is not linked to modulation of PABPN1 aggregates, but rather to myofibrillar protein homeostasis. These results indicate that reducing proteasome activity using pharmacological treatments might be beneficial for OPMD, and have important implications for the development of therapies in muscular dystrophies.

## Results

### A genome-wide genetic screen to identify suppressors of OPMD in *Drosophila*

We previously established that expression of alanine-expanded mammalian PABPN1 (PABPN1-17ala) in *Drosophila* muscles using the *UAS/Gal4* system and the muscle-specific driver *Myosin heavy chain-Gal4 (Mhc-Gal4)*, led to wing position defects resulting from affected muscle activity and progressive degeneration [40]. This hybrid model expressing the mammalian protein has proven useful to identify molecular pathways relevant to OPMD in patients [23]. To more broadly identify molecular pathways involved in OPMD, we set up a genome-wide genetic screen in *Drosophila*, based on this hybrid OPMD model. Recording defective wing position in adult flies is a long process, unsuitable for a large screen, therefore we established a lethality screen to speed up screening. Expression of *UAS-PABPN1-17ala* with the *24B-Gal4* driver that specifically expresses Gal4 in the mesoderm from embryonic stages [41], leads to larval lethality at 22˚C with only 5 to 15% of individuals surviving to pupal stage (Fig 1A). We characterized muscle defects in *UAS-PABPN1-17ala/+*; *24B-Gal4/+* individuals, using phalloidin staining to visualize Actin in late embryonic and third instar larval stages. No major defects were visible in the musculature of *UAS-PABPN1-17ala/+*; *24B-Gal4/+* late embryos (S1A Fig), showing that muscles developed normally and were maintained throughout embryogenesis following expression of PABPN1-17ala in the mesoderm. In contrast, the body-wall musculature of third instar larvae was very affected. First, 100% of *UAS-PABPN1-17ala/+*; *24B-Gal4/+* third instar larvae were paralyzed and smaller than *24B-Gal4/+* control larvae (n>100). Second, phalloidin staining of body-wall muscles revealed a generalized muscle atrophy (100% of *UAS-PABPN1-17ala/+*; *24B-Gal4/+* larval hemi-segments (n = 38) versus 0% of *24B-Gal4/+* controls (n = 28)) (S1B Fig). Third, phalloidin staining also revealed a large number of other morphological defects, including splitted, very thin and broken muscles (S1B and S1C Fig). Muscle defects in larvae resembled those described in adult OPMD muscles [40] (see below). In addition, these defects were progressive from embryonic to larval stages, as in the case of PABPN1-17ala expression in adults. Therefore, embryonic expression of PABPN1-17ala with *24B-Gal4* is a relevant model to screen for suppressors of OPMD phenotypes.

Large deficiencies from the *Drosophila* deficiency kit (Bloomington *Drosophila* Stock Center) covering most of the genome were heterozygously introduced into the *UAS-PABPN1-*

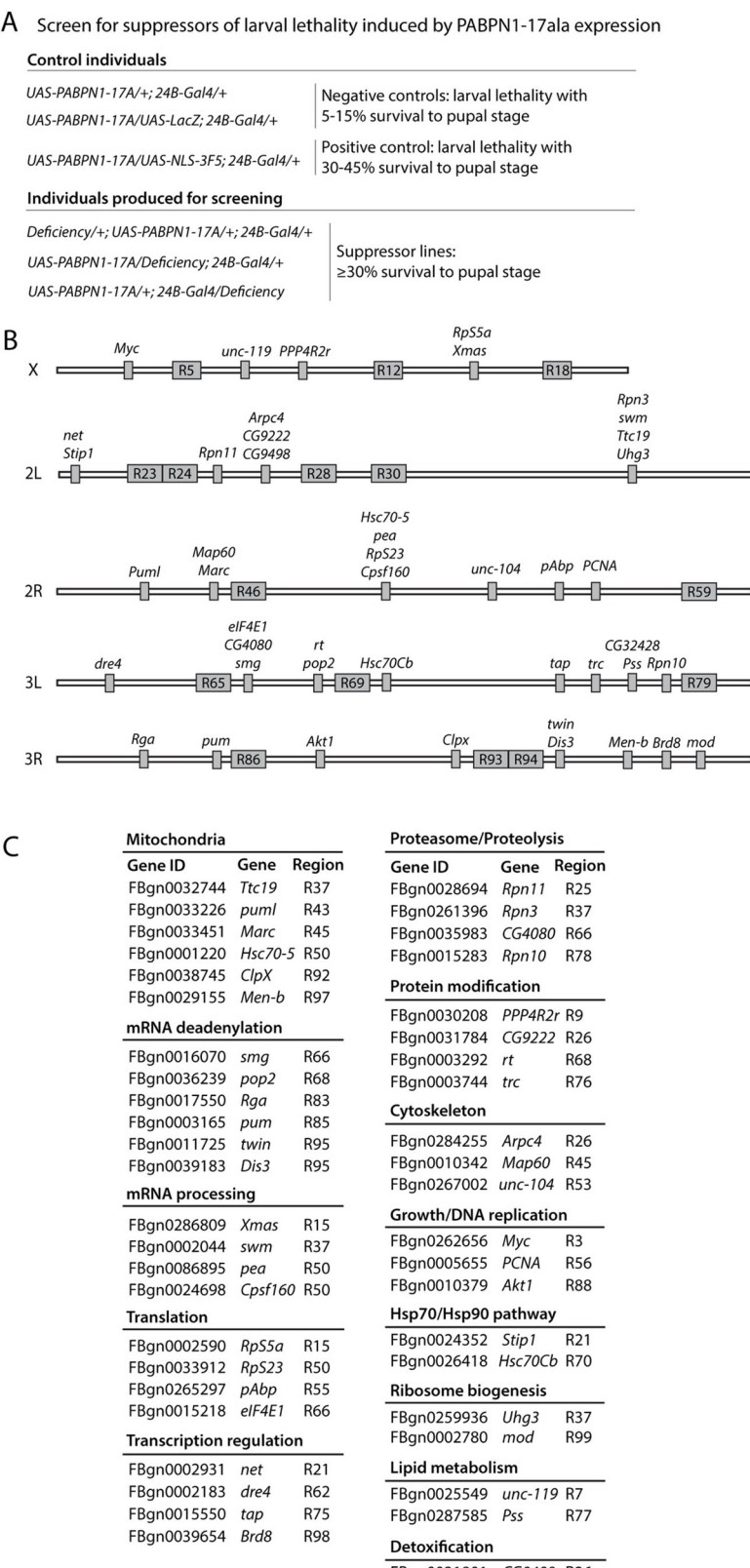

**A** Screen for suppressors of larval lethality induced by PABPN1-17ala expression

**Control individuals**

| | |
|---|---|
| *UAS-PABPN1-17A/+; 24B-Gal4/+*<br>*UAS-PABPN1-17A/UAS-LacZ; 24B-Gal4/+* | Negative controls: larval lethality with 5-15% survival to pupal stage |
| *UAS-PABPN1-17A/UAS-NLS-3F5; 24B-Gal4/+* | Positive control: larval lethality with 30-45% survival to pupal stage |

**Individuals produced for screening**

| | |
|---|---|
| *Deficiency/+; UAS-PABPN1-17A/+; 24B-Gal4/+*<br>*UAS-PABPN1-17A/Deficiency; 24B-Gal4/+*<br>*UAS-PABPN1-17A/+; 24B-Gal4/Deficiency* | Suppressor lines:<br>≥30% survival to pupal stage |

**B**

**C**

**Mitochondria**

| Gene ID | Gene | Region |
|---|---|---|
| FBgn0032744 | *Ttc19* | R37 |
| FBgn0033226 | *puml* | R43 |
| FBgn0033451 | *Marc* | R45 |
| FBgn0001220 | *Hsc70-5* | R50 |
| FBgn0038745 | *ClpX* | R92 |
| FBgn0029155 | *Men-b* | R97 |

**mRNA deadenylation**

| Gene ID | Gene | Region |
|---|---|---|
| FBgn0016070 | *smg* | R66 |
| FBgn0036239 | *pop2* | R68 |
| FBgn0017550 | *Rga* | R83 |
| FBgn0003165 | *pum* | R85 |
| FBgn0011725 | *twin* | R95 |
| FBgn0039183 | *Dis3* | R95 |

**mRNA processing**

| Gene ID | Gene | Region |
|---|---|---|
| FBgn0286809 | *Xmas* | R15 |
| FBgn0002044 | *swm* | R37 |
| FBgn0086895 | *pea* | R50 |
| FBgn0024698 | *Cpsf160* | R50 |

**Translation**

| Gene ID | Gene | Region |
|---|---|---|
| FBgn0002590 | *RpS5a* | R15 |
| FBgn0033912 | *RpS23* | R50 |
| FBgn0265297 | *pAbp* | R55 |
| FBgn0015218 | *eIF4E1* | R66 |

**Transcription regulation**

| Gene ID | Gene | Region |
|---|---|---|
| FBgn0002931 | *net* | R21 |
| FBgn0002183 | *dre4* | R62 |
| FBgn0015550 | *tap* | R75 |
| FBgn0039654 | *Brd8* | R98 |

**Proteasome/Proteolysis**

| Gene ID | Gene | Region |
|---|---|---|
| FBgn0028694 | *Rpn11* | R25 |
| FBgn0261396 | *Rpn3* | R37 |
| FBgn0035983 | *CG4080* | R66 |
| FBgn0015283 | *Rpn10* | R78 |

**Protein modification**

| Gene ID | Gene | Region |
|---|---|---|
| FBgn0030208 | *PPP4R2r* | R9 |
| FBgn0031784 | *CG9222* | R26 |
| FBgn0003292 | *rt* | R68 |
| FBgn0003744 | *trc* | R76 |

**Cytoskeleton**

| Gene ID | Gene | Region |
|---|---|---|
| FBgn0284255 | *Arpc4* | R26 |
| FBgn0010342 | *Map60* | R45 |
| FBgn0267002 | *unc-104* | R53 |

**Growth/DNA replication**

| Gene ID | Gene | Region |
|---|---|---|
| FBgn0262656 | *Myc* | R3 |
| FBgn0005655 | *PCNA* | R56 |
| FBgn0010379 | *Akt1* | R88 |

**Hsp70/Hsp90 pathway**

| Gene ID | Gene | Region |
|---|---|---|
| FBgn0024352 | *Stip1* | R21 |
| FBgn0026418 | *Hsc70Cb* | R70 |

**Ribosome biogenesis**

| Gene ID | Gene | Region |
|---|---|---|
| FBgn0259936 | *Uhg3* | R37 |
| FBgn0002780 | *mod* | R99 |

**Lipid metabolism**

| Gene ID | Gene | Region |
|---|---|---|
| FBgn0025549 | *unc-119* | R7 |
| FBgn0287585 | *Pss* | R77 |

**Detoxification**

| Gene ID | Gene | Region |
|---|---|---|
| FBgn0031801 | *CG9498* | R26 |

**unclassified**

| Gene ID | Gene | Region |
|---|---|---|
| FBgn0052428 | *CG32428* | R77 |

**Fig 1. Overview of the genetic screen to identify suppressors of larval lethality induced by expression of PABPN1-17ala in the mesoderm.** (A) Genotypes of individuals generated for screening and for negative and positive controls. Deficiencies leading to at least 30% of individuals surviving to pupal stage were considered to be suppressor. (B) Genes and suppressive regions identified in the screen. Each chromosome arm (X, 2L, 2R, 3L, 3R) is depicted with suppressor genes and suppressive regions (R). Suppressive regions are regions identified using deficiencies and for which the specific suppressor genes could not be identified following the screen pipeline. (C) List of suppressor genes identified in the screen classified using Gene Ontology. Mutant alleles are listed in S1 Table.

*17ala/+*; *24B-Gal4/+* background, and deficiencies leading to at least 30% of individuals surviving to pupal stage were considered as suppressor (Fig 1A). This threshold was determined using a positive control line *UAS-NLS-3F5* expressing an anti-PABPN1 intrabody that we have shown to be very efficient in preventing OPMD defects in *Drosophila* [31] (Fig 1A). In total 252 large deficiencies were screened covering about 87% of the genome and 63 deficiencies were found to be suppressor (S2A Fig). Suppressor genes within these deficiencies were identified as follows. For each large deficiency found to be suppressor, the same screen was applied with smaller deficiencies spanning the large deficiency to identify smaller suppressive regions (S1 Table). 92 smaller deficiencies were screened allowing to narrow down 27 suppressive genomic regions (S1 Table and S2B Fig). Then, the same screen was applied again with heterozygous mutants or RNAi of candidate genes localized in suppressive regions (S1 Table). Candidate genes were determined based on their reported relationships with OPMD and/or PABPN1, their higher expression in OPMD *Drosophila* muscles compared to wild type [31], and with the help of Endeavour-HighFly, a software application for the computational prioritization of candidate genes [42]. In total, 142 mutant genes were tested allowing to identify 46 suppressor genes of OPMD phenotypes in *Drosophila* and pinpoint several molecular pathways involved in OPMD (Fig 1B and 1C and S1 Table). Importantly, among these pathways, a number have previously been shown to contribute to OPMD. Although this was not unexpected since a number of candidate genes to be analyzed were selected based on their previous links with OPMD, this nonetheless validated the screen. The most prominent pathways (i.e. with the higher number of genes): "Mitochodrion" and "mRNA deadenylation" (Fig 1C) were previously demonstrated to play a key role in OPMD pathogenesis [23]. "mRNA processing" and "Translation" were other identified pathways consistent with the molecular functions of PABPN1. The "Hsp70/Hsp90" pathway that came up in the screen was also previously shown to contribute to OPMD [40]. Finally, another notable pathway that was identified by several genes was "Proteasome/Proteolysis" that we also previously reported to be highly deregulated in OPMD animal models and patients [37].

Overall, the genome-wide genetic screen based on survival of lethal expression of PABPN1-17ala allowed to identify a large number of new genes and pathways contributing to OPMD pathogenesis.

## Functional implication of the UPS in OPMD

Although the UPS was shown to be consistently deregulated in OPMD animal models and in patients [37], its functional implication in OPMD has not been addressed *in vivo*. Consistent with upregulation of a number of UPS components in OPMD during disease progression [23,37], results of the genetic screen suggested that decreasing the gene dosage of UPS components might be beneficial for OPMD in *Drosophila* (Fig 1C). Strikingly, mutants of three subunits of the proteasome, Rpn3, Rpn10 and Rpn11 were found to reduce OPMD phenotypes. To address more thoroughly the functional involvement of the UPS in OPMD, we performed a second screen based again on the rescue of *UAS-PABPN1-17ala/+; 24B-Gal4/+* larval lethality, but targeted on UPS components. Heterozygous mutants of UPS components were

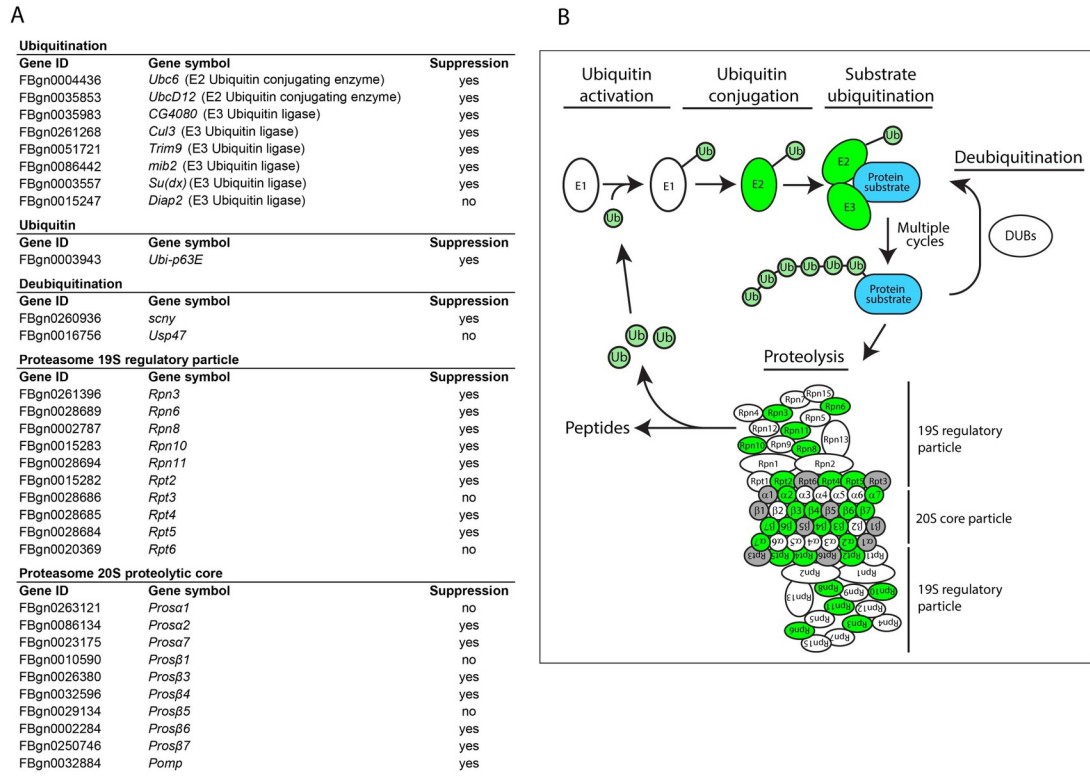

**A**

**Ubiquitination**

| Gene ID | Gene symbol | Suppression |
|---|---|---|
| FBgn0004436 | *Ubc6* (E2 Ubiquitin conjugating enzyme) | yes |
| FBgn0035853 | *UbcD12* (E2 Ubiquitin conjugating enzyme) | yes |
| FBgn0035983 | *CG4080* (E3 Ubiquitin ligase) | yes |
| FBgn0261268 | *Cul3* (E3 Ubiquitin ligase) | yes |
| FBgn0051721 | *Trim9* (E3 Ubiquitin ligase) | yes |
| FBgn0086442 | *mib2* (E3 Ubiquitin ligase) | yes |
| FBgn0003557 | *Su(dx)* (E3 Ubiquitin ligase) | yes |
| FBgn0015247 | *Diap2* (E3 Ubiquitin ligase) | no |

**Ubiquitin**

| Gene ID | Gene symbol | Suppression |
|---|---|---|
| FBgn0003943 | *Ubi-p63E* | yes |

**Deubiquitination**

| Gene ID | Gene symbol | Suppression |
|---|---|---|
| FBgn0260936 | *scny* | yes |
| FBgn0016756 | *Usp47* | no |

**Proteasome 19S regulatory particle**

| Gene ID | Gene symbol | Suppression |
|---|---|---|
| FBgn0261396 | *Rpn3* | yes |
| FBgn0028689 | *Rpn6* | yes |
| FBgn0002787 | *Rpn8* | yes |
| FBgn0015283 | *Rpn10* | yes |
| FBgn0028694 | *Rpn11* | yes |
| FBgn0015282 | *Rpt2* | yes |
| FBgn0028686 | *Rpt3* | no |
| FBgn0028685 | *Rpt4* | yes |
| FBgn0028684 | *Rpt5* | yes |
| FBgn0020369 | *Rpt6* | no |

**Proteasome 20S proteolytic core**

| Gene ID | Gene symbol | Suppression |
|---|---|---|
| FBgn0263121 | *Prosα1* | no |
| FBgn0086134 | *Prosα2* | yes |
| FBgn0023175 | *Prosα7* | yes |
| FBgn0010590 | *Prosβ1* | no |
| FBgn0026380 | *Prosβ3* | yes |
| FBgn0032596 | *Prosβ4* | yes |
| FBgn0029134 | *Prosβ5* | no |
| FBgn0002284 | *Prosβ6* | yes |
| FBgn0250746 | *Prosβ7* | yes |
| FBgn0032884 | *Pomp* | yes |

**B**

**Fig 2. Results of the genetic screen targeted towards the UPS to identify suppressors of larval lethality induced by expression of PABPN1-17ala in the mesoderm.** (A) Results of the UPS screen. Mutant alleles are listed in S2 Table. They were used in the screen as heterozygote, as indicated in Fig 1A. (B) Schematic representation of the UPS. UPS components that were found to be positive in the screen are depicted in green. Subunits of the proteasome that were tested in the screen and had no suppressor effect are depicted in grey.

introduced into the *UAS-PABPN1-17ala/+; 24B-Gal4/+* background, and survival to pupal stage was recorded. In total 31 components of the UPS were tested, including E2 ubiquitin conjugating-enzymes, E3 ubiquitin ligases, deubiquitinating enzymes and subunits of both the regulatory and core particles of the proteasome (Fig 2A and 2B). 24 of the 31 (77%) tested genes were found to be suppressors of OPMD and the positive genes fell into both the ubiquitination and degradation steps of protein processing by the UPS (Fig 2A and 2B and S2 Table). Deubiquitinating enzymes that act oppositely to ubiquitin ligases were not expected to be identified as OPMD suppressors. Nevertheless, the *scny* gene that encodes the Usp36 deubiquitinating enzyme was recorded as suppressor (Fig 2A). This might reflect its recently reported role in stabilizing Myc protein, another suppressor of OPMD (Fig 1C), and in regulating an E3 ubiquitin ligase [43].

These results reveal a functional role of the UPS in OPMD pathogenesis and indicate that decreasing UPS activity is beneficial for OPMD in the *Drosophila* model.

## UPS mutants reduce OPMD muscle weakness and degeneration

To further understand the role of UPS in OPMD pathogenesis, we selected five UPS components and analyzed the effects of mutants on adult muscles. Rpn10 and Rpn11 are subunits of the 19S regulatory particle of the proteasome. Prosβ4 is a subunit of the proteasome 20S core particle. The proteasome maturation protein, Pomp coordinates the assembly of the 20S core proteasome complex and is required for production of newly formed proteasome [44–47].

Mind bomb 2 (Mib2) is a E3 ubiquitin ligase specific to muscles whose E3-RING-finger domains are required for myoblast fusion in the embryo, and development of thoracic indirect flight muscles (IFMs) in adult flies [48,49]. In addition, although Mib2 direct targets have not been identified, Mib2 was shown to physically interact with the Z-band-localized nonmuscle myosin [48]. Mutants of these five UPS components were selected to analyze their effect on OPMD phenotypes in adults. All mutants were homozygous lethal. We checked using RT-qPCR that the levels of the corresponding mRNA were reduced in heterozygous mutant flies (S3A and S3B Fig). This was the case for all mutants except for $Rpn10^{G6601}$. However, this $Rpn10$ mutant allele contains a $P$-element insertion in the coding sequence just downstream of the start codon (S3A Fig). Therefore, although $Rpn10$ mRNA levels were not decreased, the coding capacity is expected to be strongly affected in this mutant. We used the $Act88F$-$PABPN1$-$17ala$ stock that we previously described [23] as an OPMD model in adults. This stock specifically expresses alanine-expanded PABPN1 in adult IFMs, leading to 50–60% of flies with abnomal wing posture at day 6 of adulthood (Fig 3A and 3B) [23]. We previously reported that wing posture defects (wings held up or down) in OPMD fly models reflected both affected muscle function prior to the appearance of altered sarcomeric structure, and muscle degeneration [40]. The $Act88F$-$PABPN1$-$17ala$ stock was crossed with mutants of the five selected UPS components. The presence of all five heterozygous mutants significantly reduced the percentage of flies with wing posture defects to between 34 and 6% (Fig 3B). Because the UPS mutants affect all tissues, we used the $UAS/Gal4$ system to validate the muscle-specificity of their suppressor effect by knocking-down $Pomp$ in muscles. Reduced expression of $Pomp$ in $Act88F$-$PABPN1$-$17ala/+$ flies, following activation of $UAS$-$Pomp$-$RNAi$ with the muscle-specific driver $Mhc$-$Gal4$, decreased the percentage of flies with defective wing posture (S3C Fig).

To better evaluate the predictive value of the larval lethality screen on adult OPMD defects at the level of individual genes, we tested the suppressor effect in adults of the $Pros\beta1$ heterozygous mutant that scored negative in the lethality screen (Fig 2A and S2 Table). $Pros\beta1$ heterozygous mutant reduced the percentage of $Act88F$-$PABPN1$-$17ala/+$ flies with wing posture defects, revealing a suppressor effect of this mutant in adults (S3D Fig). This suggested that the number of UPS components having a suppressor effect on OPMD phenotypes based on the larval lethality screen was yet underestimated.

OPMD defects are progressive and the percentage of $Act88F$-$PABPN1$-$17ala/+$ flies with abnormal wing posture increased to 81% at day 11 (Fig 3B). The significant rescue of wing posture defects in presence of the five UPS component mutants was maintained at day 11 (Fig 3B).

We have previously reported that expression of PABPN1-17ala in thoracic muscles induces progressive muscle degeneration that can be recorded by visualizing IFMs under polarized light [30,40]. These muscles are composed of six dorso-longitudinal muscles (DLMs) involved in downward wing movements, and seven dorso-ventral muscles (DVMs) involved in upward wing movements (Fig 3C). We analyzed muscle degeneration in $Act88F$-$PABPN1$-$17ala/+$ flies at day 11 in the absence or presence of UPS heterozygous mutants by counting the number of affected DLMs and DVMs per thorax. In the absence of UPS mutants, 78% of DLMs and 50% of DVMs were altered in PABPN1-17ala expressing flies at day 11 (Fig 3C and 3D). In the presence of heterozygous mutants for the five UPS components, the percentages of both affected DLMs and DVMs were significantly reduced, with a stronger effect for DLMs (Figs 3D and S4A).

We then studied OPMD muscle defects and rescue with UPS mutants at the level of sarcomere organization using immunostaining to visualize Mhc and Kettin (also known as Titin), a large Actin-binding protein that accumulates in Z discs. Immunostaining of adult IFMs revealed a series of defects in $Act88F$-$PABPN1$-$17ala/+$ flies related to defects observed upon

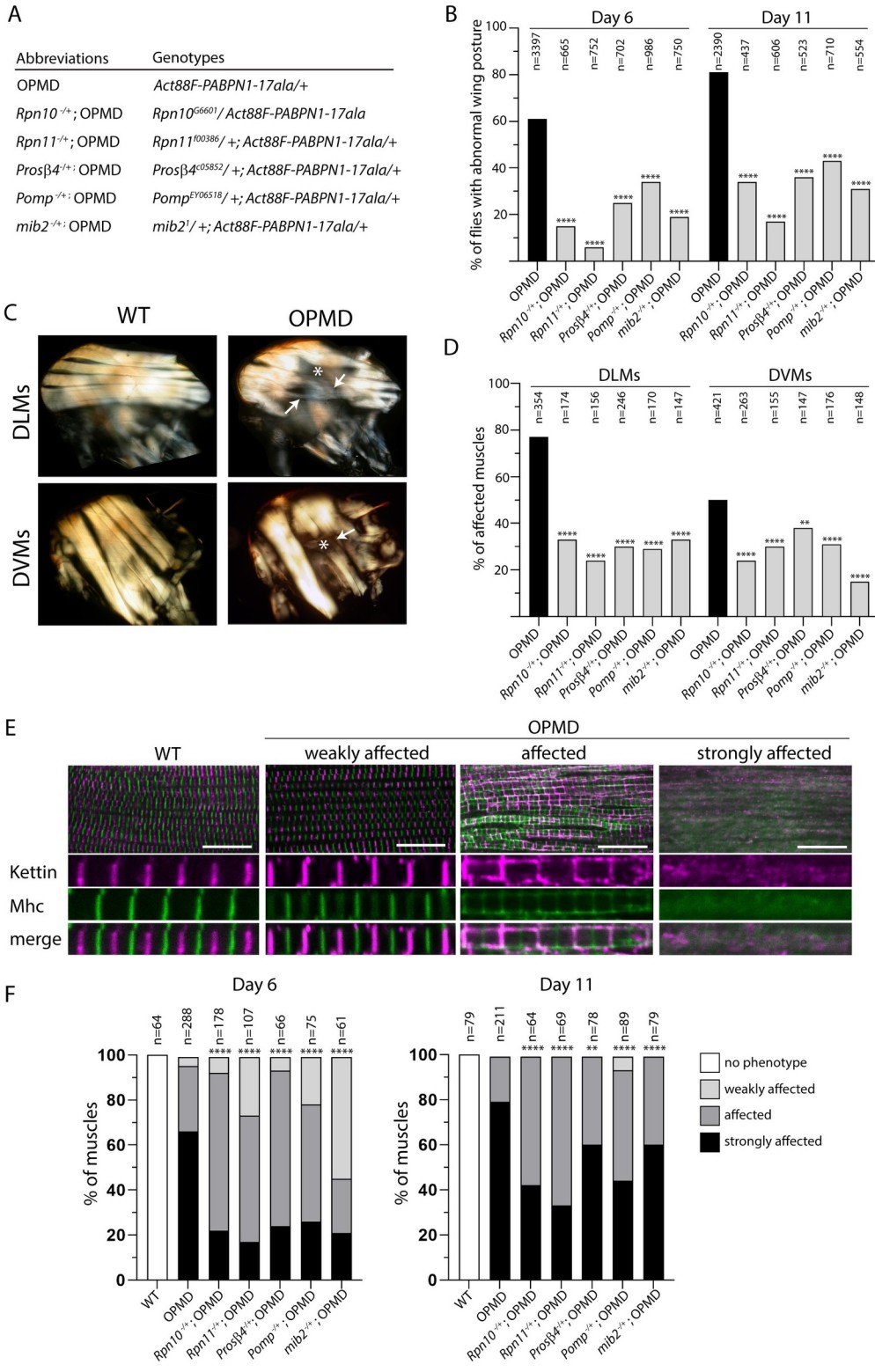

**Fig 3. Decrease of muscle weakness and degeneration with mutants of UPS components.** (A) Genotypes and abbreviations used in Figs 3–6. (B) Percentages of flies with abnormal wing position were scored at day 6 or day 11 of adulthood at 25°C. Flies were scored from three to five independent crosses; the numbers of scored flies are indicated (n). **** $p$-value <0.0001, using the $\chi_2$ test. (C) Indirect flight muscles in wild-type and OPMD adult thoraxes visualized under polarized light at day 11. Six dorso-longitudinal muscles (DLMs) and seven dorso-ventral muscles

(DVMs) were visualized and scored per hemi-thorax. White arrows and stars indicate very thin and broken or degenerating muscles, respectively, in OPMD fly thoraxes. (D) Quantification of affected muscles (as shown in the OPMD panel in C) in the different genotypes. The numbers of scored DLMs or DVMs are indicated (n). ****p-value <0.0001, **p-value <0.01, using the $\chi_2$ test. (E) Immunostaining of wild-type and OPMD DLMs with anti-Kettin (magenta) and anti-Mhc (green) to visualize myofibrils and sarcomeric structure. The three categories of defects are exemplified from OPMD DLMs. Scale bar: 15 μm. (F) Quantification of wild-type and affected DLMs in the three categories in the different genotypes at days 6 and 11. The numbers of scored muscles are indicated (n). ****p-value <0.0001, **p-value <0.01, using the Fisher's exact test.

knock-down of genes encoding myofibrillar components or involved in muscle organization [50]. We classified these defects under three categories: weakly affected, affected and strongly affected muscles (Fig 3E). The "weakly affected" phenotype corresponded to a slight mislocalization of Mhc to Z bands, and Kettin outside Z bands. In "affected" muscles, the sarcomeric structure was still visible, however, both Mhc and Kettin were highly delocalized, in Z bands and at the sarcomere periphery for Mhc, and outside Z bands in aggregates and at the sarcomere periphery for Kettin. In addition, myofibrils were thiner suggesting atrophy. The "strongly affected" phenotype corresponded to irregular or degenerated myofibrils and a complete loss of sarcomeric organization. Quantification of muscles with these three phenotype categories at day 6 and day 11 of adulthood confirmed the progressivity of muscle defects in OPMD flies with a higher percentage of muscles being strongly affected at day 11 and no muscles in the weakly affected category (Fig 3F). Muscle quantification in the presence of UPS heterozygous mutants revealed that all five mutants significantly reduced muscle defects at day 6, leading to lower percentages of strongly affected muscles and increased percentages of weaker phenotypes. This positive effect was still visible at day 11 although to a lower extent.

Because OPMD defects in *Drosophila* depends on the levels of PABPN1-17ala, we verified using western blots, that the reduced wing position defects and muscle degeneration in the presence of UPS heterozygous mutants were not due to decreased levels of PABPN1-17ala (S4B Fig).

Taken together, these results reveal that the UPS plays a key role in muscle defects in OPMD.

## Increased proteasome activity in OPMD muscles

The overexpression of UPS components at the transcriptomic level in OPMD muscles [37], as well as the genetic rescue of OPMD muscle defects in the presence of UPS heterozygous mutants strongly suggest an increase of proteasome level and activity in OPMD muscles. We confirmed this hypothesis by directly measuring the chymotrypsin-like activity of the proteasome using *in vitro* assays. The chymotrypsin-like activity that is a proxy for proteasome levels *in vitro*, was significantly increased by approximately two-fold in *Act88F-PABPN1-17ala/+* thoracic muscles compared to wild type muscles at both day 3 and day 6 (Fig 4A). Consistent with this, in gel assays to visualize proteasome assembly and activity, revealed higher level of active 26S proteasome in OPMD thoracic muscles compared to wild type, at day 3 and day 6 (Fig 4B). Furthermore, western blots also uncovered increased amounts of proteasome subunits in OPMD thoracic muscles at both day 3 and day 6 (S5A Fig). Measurements of the chymotrypsin-like activity in OPMD muscles in the presence of heterozygous mutants of proteasome subunits showed that it was reduced to wild-type levels by the *Rpn10* mutation, although this reduction was not observed with mutants of the other proteasome subunits, Rpn11 and Prosβ4 (Fig 4A). This different outcome with the *Rpn10* mutant might result from the specific role of this subunit in promoting 26S proteasome stability and thus proteasome disassembly upon reduction of Rpn10 levels [51–53]. When chymotrypsin-like activity was measured in thoracic muscles with the same heterozygous mutants but independently of PABPN1-17ala expression, a decrease was also observed with the *Rpn10* mutant only

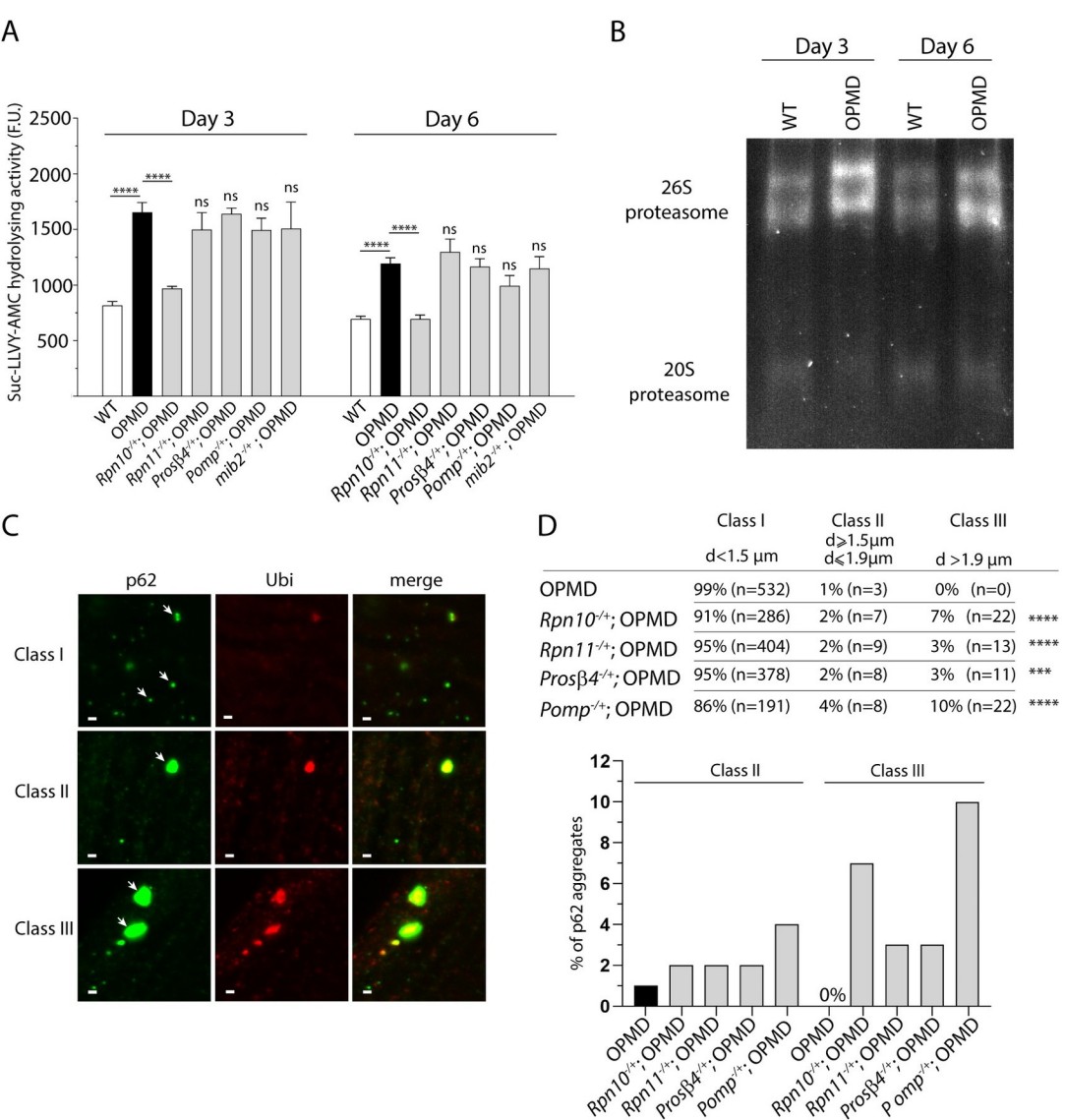

**Fig 4. Proteasome activity is higher in OPMD than wild-type thoracic muscles.** (A) The proteasome chymotrypsin-like activity in protein extracts from thoracic muscles was measured using hydrolysis of the N-succinyl-leucine-leucine-valine-tyrosine-7-amino-4-methylcoumarin (Suc-LLVY-AMC) through AMC fluorescence, at day 3 and day 6 of adulthood. Quantifications were from wild-type, OPMD (*Act88F-PABPN1-17ala/+*) and OPMD with heterozygous UPS mutant thoracic muscles. Means of three biological replicates quantified three times. Error bars represent SEM. \*\*\*\**p*-value <0.0001, ns: non significant compared to OPMD, using the unpaired Student's *t*-test. F.U.: fluorescent units. (B) In-gel assay of proteasome assembly and activity. Proteasome complexes were separated using native gel electrophoresis. The gel was then incubated in the proteasome substrate Suc-LLVY-AMC and proteasome activity was visualized by the release of fluorescent AMC. (C) Immunostaining of IFMs with anti-p62 (green) and anti-ubiquitin (red) at day 3. White arrows point to each class of aggregates. Class I, II and III aggregates are exemplified from OPMD, *Rpn10*-/+; OPMD and *Pomp*-/+; OPMD IFMs, respectively. Note that small p62 foci do not systematically colocalize with ubiquitin whereas large aggregates do. Scale bar: 1 μm. (D) Quantification of the three classes of p62 aggregates in the different genotypes at day 3 of adulthood. \*\*\*\**p*-value <0.0001, \*\*\**p*-value <0.001, using the Fisher's exact test.

(S5B Fig). Finally, as expected the chymotrypsin-like activity was not affected by heterozygous mutation of *mib2* that encodes an E3-ubiquitin ligase (Figs 4A and S5B).

Although quantification of the chymotrypsin-like activity *in vitro* can inform on active proteasome levels, it does not provide an accurate quantification of proteasome activity *in vivo*.

Specifically, this *in vitro* assay is based on peptide degradation independently of ubiquitination and does not fully assess the complex process that leads to the degradation of a protein substrate by the proteasome *in vivo* [54]. We, therefore, used a previously reported *in vivo* assay to evaluate proteasome activity in OPMD thoracic muscles in the presence or absence of proteasome subunit heterozygous mutants. The autophagy receptor p62 binds ubiquitin and forms aggregates with ubiquitinated proteins upon accumulation of these proteins [55,56]. In particular, p62 was shown to form large ubiquitin-containing aggregates in the *Drosophila* fat body when proteasome activity was impaired through downregulation of various proteasome subunits [57]. We used this assay and quantified p62-ubiquitin aggregates using immunostaining of OPMD IFMs either or not heterozygous mutant for the proteasome subunits Rpn10, Rpn11 and Prosβ4, or the Pomp proteasome chaperone. Large p62-ubiquitin aggregates (class III: diameter >1.9 μm and reaching up to 4 μm) formed in the presence of all heterozygous mutants, whereas they were not found in OPMD muscles alone. In addition, the number of intermediate aggregates (class II: diameter 1.5 to 1.9 μm) increased in the presence of heterozygous mutants compared to OPMD alone (Fig 4C and 4D). These data are consitent with impaired proteasome activity in heterozygous mutants of proteasome subunits and Pomp.

We conclude that OPMD is associated with increased proteasomal content and activity in muscles that substantially contributes to OPMD defects.

## Reduced OPMD muscle defects by UPS mutants do not correlate with levels of PABPN1 aggregation

Deregulation of the UPS has previously been proposed to contribute to OPMD through the modulation of PABPN1-17ala aggregation [35–37]. However, this hypothesis has been tested in cell models of OPMD, but not *in vivo*, in animal models. Furthermore, nuclear aggregates of alanine-expanded PABPN1 in cell cultures do not display the dense fibrillar structure of those in OPMD patients and animal models [7,25,40,58], indicating that PABPN1 aggregates in cell models do not accurately reproduce aggregates in OPMD patients. To address whether the UPS mutants reduced degeneration of OPMD muscle through altering PABPN1-17ala aggregation, we analyzed PABPN1 nuclear aggregates in *Act88F-PABPN1-17ala/+* thoracic muscles in the absence or presence of the UPS heterozygous mutants at day 11. We found that *Rpn11*, *Prosβ4* and *Pomp* heterozygous mutants reduced PABPN1-17ala aggregation. For all three mutants, the number of nuclei with an aggregate was significantly lower, and the size of aggregates was also reduced in the presence of *Pomp*$^{-/+}$ (Fig 5A–5D). In contrast, *Rpn10* heterozygous mutants increased PABPN1-17ala aggregation; indeed the size of aggregates were bigger in the presence of this mutant (Fig 5C and 5D). Enhanced PABPN1 aggregation in the *Rpn10* mutant background was consistent with reduced proteasomal activity measured *in vitro* using the chymotrypsin-like activity (Fig 4A). Finally, PABPN1-17ala aggregation was not affected by *mib2* heterozygous mutant (Fig 5B–5D), in agreement with the proposed link of this E3 ubiquitin ligase with proteins involved in sarcomeric structure, rather than PABPN1.

Strikingly, muscle degeneration was reduced in the presence of *Rpn10* heterozygous mutant (Fig 3B–3D) despite increased PABPN1 aggregation. These data show that the reduction of muscle degeneration in the presence of UPS mutants can be uncoupled from reduced PABPN1 aggregation.

## Myofibrillar proteins are more ubiquitinated and degraded in OPMD muscles

The UPS is a key regulator of muscle mass. Since the implication of the UPS in OPMD pathogenesis could be uncoupled from PABPN1 aggregation, we analyzed the UPS contribution to the degradation of myofibrillar proteins. First, the levels of ubiquitinated proteins in wild type

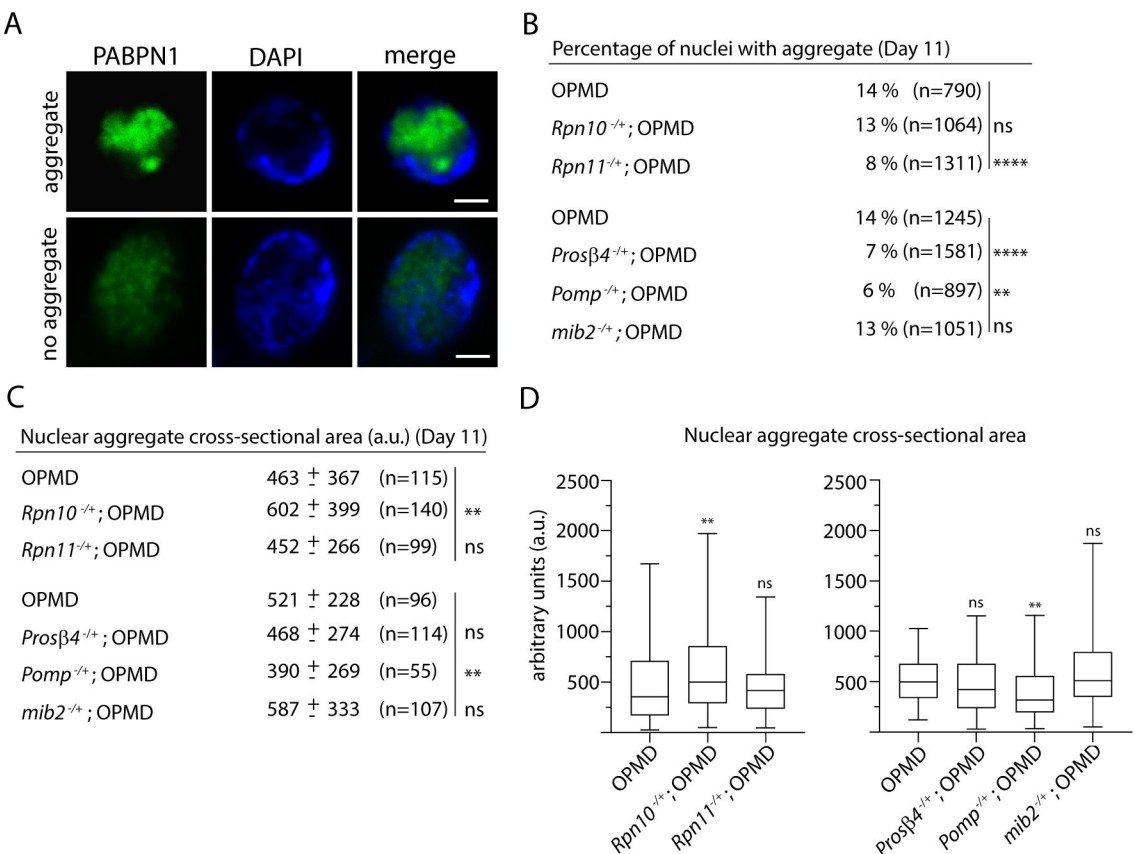

**Fig 5. PABPN1 aggregation in the presence of UPS hererozygous mutants does not correlate with OPMD muscle defects.** (A) Example of a PABPN1 nuclear aggregate. Immunostaining of IFMs from OPMD (*Act88F-PABPN1-17ala/+*) adult flies with anti-PABPN1 (green), showing nuclei with and without a nuclear aggregate. DNA was visualized with DAPI (blue). Scale bar: 2 μm. (B) Percentages of nuclei with PABPN1 nuclear aggregates in OPMD IFMs at day 11. Immunostaining of thoracic muscles with anti-PABPN1 and DAPI were used to visualize and score aggregates. The number of scored nuclei is indicated (n). ****$p$-value $<0.0001$, **$p$-value $<0.01$, ns: non significant, using the $\chi_2$ test. (C, D) Quantification of PABPN1 nuclear aggregate areas. Each aggregate was delimited in a focal plan and the surface was calculated using ImageJ. (C) Mean values of areas with SD are indicated in arbitrary units. The number of scored aggregates is indicated (n). (D) The distribution of cross-sectional areas is presented as box plots. The boxes represent 50% of the values; the horizontal lines within the boxes correspond to medians. **$p$-value $<0.01$, ns: non significant, using the unpaired Student's $t$-test. Quantifications in B-D were from two independent experiments.

and OPMD thoracic muscles were analyzed by western blots and found to be significantly higher in OPMD muscles at day 3 and day 6 of adulthood (Fig 6A). To specifically assess ubiquitination of myofibrillar proteins, western blots were performed using extracts enriched in myofibrillar proteins from wild type and OPMD thoracic muscles. Myofibrillar proteins were more ubiquitinated in OPMD muscles at days 3, 6 and 11 (Fig 6B).

We then directly quantified the levels of the contractile proteins Actin and Mhc in wild type and OPMD thoracic muscles at days 3, 6 and 11. Both Actin and Mhc amounts were reduced in OPMD muscles compared to wild type at the three time points (Fig 6C and 6D). Importantly, Actin and Mhc degradation was recorded at day 3, when muscles were only slightly affected (S6 Fig), revealing that myofibrillar protein degradation preceded strong muscle degeneration. To address the functional importance of this degradation of myofibrillar proteins in OPMD pathogenesis, we addressed whether myofibrillar protein levels were restored in OPMD thoracic muscles at day 3, in the presence of heterozygous UPS mutants. Strikingly, both Actin and Mhc levels showed a strong tendency to increase in OPMD muscles in the presence of all UPS mutants, although this increase was not statistically significant for all

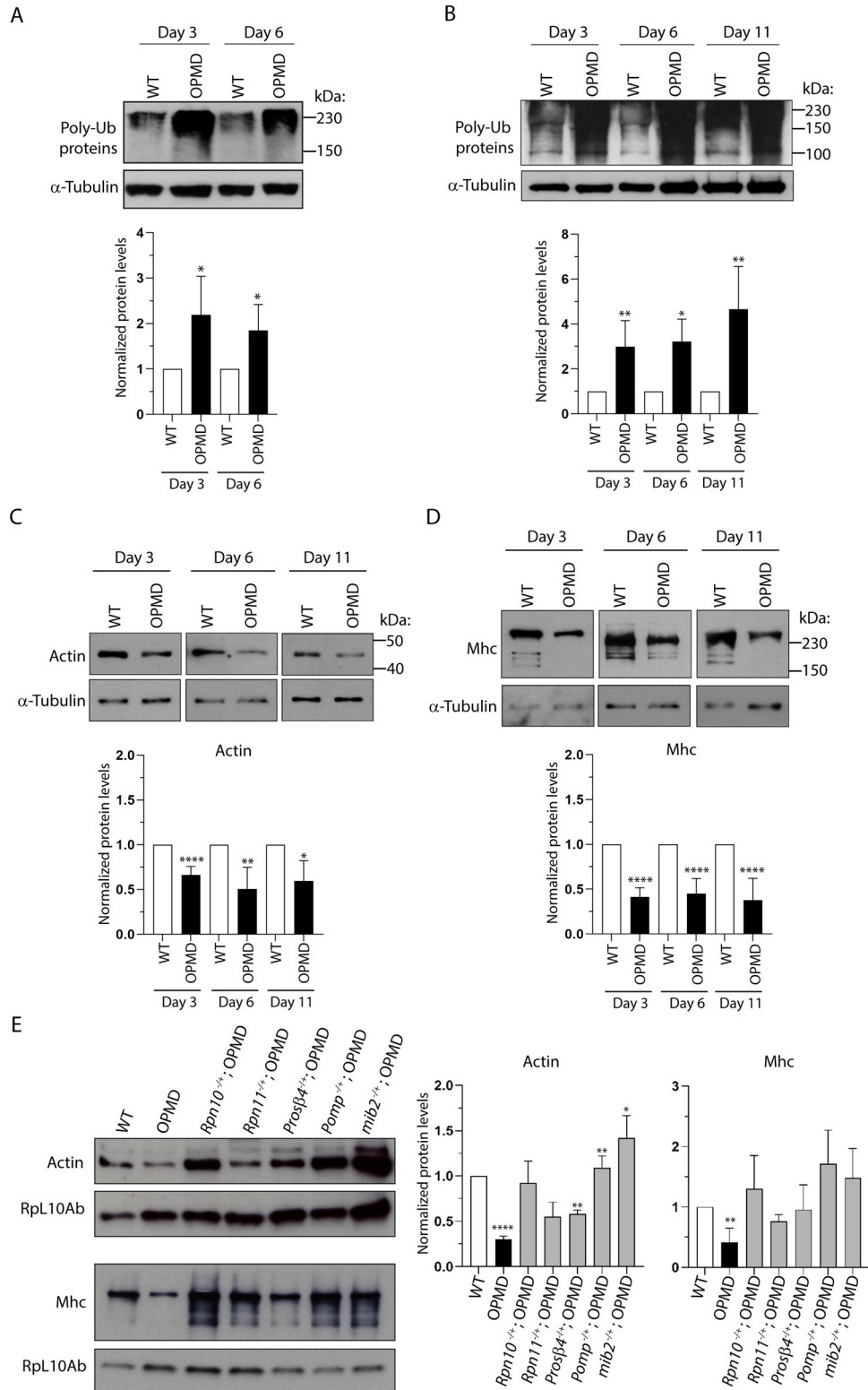

**Fig 6. Myofibrillar proteins are more ubiquitinated and degraded in OPMD than in wild-type muscles.** (A) Western blot of protein extracts from thoracic muscles of wild-type and OPMD (*Act88F-PABPN1-17ala/+*) flies at days 3 and 6 revealed with anti-ubiquitin, showing the levels of polyubiquitinated proteins. α-Tubulin was used as a loading control. Quantification was performed using the ImageJ sofware with four biological replicates. Error bars represent SD. *$p$-value <0.05 using the unpaired Student's $t$-test. (B) Western blot of myofibrillar proteins extracted

from thoracic muscles of wild-type and OPMD (*Act88F-PABPN1-17ala/+*) flies at days 3, 6 and 11 revealed with anti-ubiquitin, showing the levels of polyubiquitinated proteins. α-Tubulin was used as a loading control. Quantification was performed using the ImageJ sofware with three to five biological replicates. Error bars represent SD. **$p$-value $<$0.01, *$p$-value $<$0.05, using the unpaired Student's *t*-test. (C, D) Western blots of myofibrillar proteins extracted from thoracic muscles of wild-type and OPMD (*Act88F-PABPN1-17ala/+*) flies at days 3, 6 and 11 revealed with (C) anti-Actin or (D) anti-Mhc, showing the lower levels of Actin and Mhc in OPMD muscles. α-Tubulin was used as a loading control. Quantification was performed using ImageJ with four to eight biological replicates. Error bars represent SD. ****$p$-value $<$0.0001, **$p$-value $<$0.01, *$p$-value $<$0.05, using the unpaired Student's *t*-test. (E) Western blots of myofibrillar proteins extracted from thoracic muscles of wild-type, OPMD, and OPMD with heterozygous UPS mutant flies at day 3, revealed with anti-Actin (top panel) and anti-Mhc (bottom panel). Genotypes are indicated in Fig 3A. The ribosomal protein RpL10Ab was used as a loading control. Quantification was performed using ImageJ with three biological replicates. Error bars represent SD. ****$p$-value $<$0.0001, **$p$-value $<$0.01, *$p$-value $<$0.05 using the unpaired Student's *t*-test. OPMD was compared to wild type (WT), whereas the OPMD with heterozygous UPS mutant genotypes were compared to OPMD; the lack of stars for these genotypes indicates a lack of statistical significance.

mutants due to some variability (Fig 6E). Actin and Mhc levels could reach wild-type levels in OPMD muscles in the presence of *Rpn10*, *Pomp* and *mib2* heterozygous mutants.

These results are consistent with an important role of the UPS in myofibrillar protein degradation in OPMD muscle defects and indicate that the implication of the UPS in OPMD pathogenesis involves its function in the degradation of myofibrillar proteins.

## Pharmacological inhibition of proteasomal activity reduces OPMD muscle defects

To confirm the importance of the UPS in OPMD, we tested the effect of oral treatment with the inhibitor of proteasomal activity, MG132. The drug or the DMSO solvent alone were provided in the food from larval stages. First, the toxicity of MG132 was analyzed using wild-type larvae by recording the percentage of survival to adulthood of larvae fed with drug-supplemented food. MG132 was found to be non-toxic up to 600 μM in the food (Fig 7A). The drug was then provided at the increasing concentrations of 400, 500 and 600 μM to *Act88F-PABPN1-17ala/+* individuals from larval stage up to the day of recording at adulthood. Wing posture defects were quantified from day 3 to day 11. Oral treatment with MG132 compared with DMSO alone, significantly reduced wing position defects at all time points and with the three concentrations, with a slightly stronger effect at 600 μM (Fig 7B). We then analyzed the effect of 600 μM of MG132 on PABPN1-17ala aggregation. Strikingly, MG132 treatment led to an increase of PABPN1-17ala aggregation that was visible through an increase of both the percentage of nuclei having an aggregate (15% at day 11 following MG132 treatment versus 11% in the presence of DMSO alone) (Fig 7C), and the size of aggregates (Fig 7D and 7E). We verified using western blots that the level of PABPN1-17ala was not affected by the MG132 treatment (Fig 7F). The enhancement of PABPN1 aggregation in the presence of MG132 was consistent with the reported regulation of PABPN1-17ala aggregation by the UPS in cell cultures [35,36]. Importantly, these *in vivo* data allowed to uncouple the UPS effect on PABPN1-17ala aggregation and its role in muscle degeneration in OPMD. Although, as expected PABPN1 aggregation load was higher upon reduction of proteasomal activity, defects in wing position that reflect muscle weakness and degeneration were decreased.

These results strongly reinforce the notion that the contribution of the UPS to OPMD pathogenesis is independent of the level of PABPN1-17ala aggregation, but depends instead on its increased activity against muscle proteins. Moreover, they provide a proof-of-concept that pharmacological treatment reducing proteasomal activity might be beneficial for OPMD.

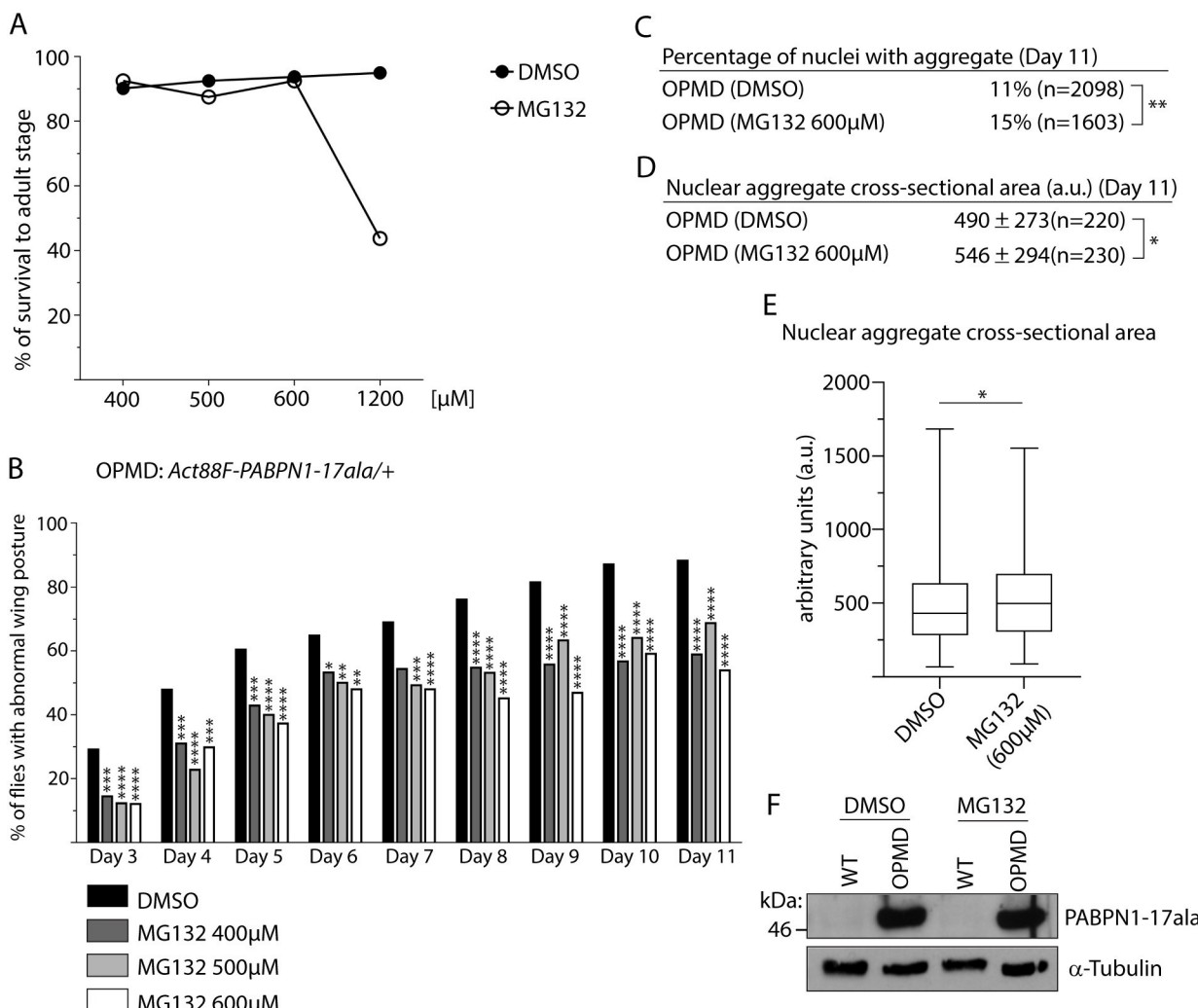

**Fig 7. Pharmacological inhibition of the proteasome with MG132 reduces OPMD muscle defects.** (A) Toxicity of increasing concentrations of MG132 in DMSO or DMSO alone provided in the food to wild-type larvae. The percentage of flies reaching adulthood is represented. For each concentration, 240 individuals from three independent experiments were scored. (B) Quantification of wing posture defects of OPMD (*Act88F-PABPN1-17ala/+*) flies following oral treatment with DMSO alone or MG132 at increasing concentrations from first instar larval stage. Wing position defects were scored every day from day 3 to day 11 of adulthood. The numbers of scored flies from two independent experiments were: 299 to 383 for 0.2% DMSO; 103 to 142 for 400 μM MG132; 100 to 135 for 500 μM MG132; 83 to 137 for 600 μM MG132. ****$p$-value <0.0001, ***$p$-value <0.001, **$p$-value <0.01, *$p$-value <0.05, using the $\chi_2$ test. (C) Percentages of nuclei with PABPN1 nuclear aggregates in OPMD IFMs at day 11 of adulthood, following oral treatment from first instar laval stage with DMSO alone or MG132 at 600 μM. Immunostaining of thoracic muscles with anti-PABPN1 and DAPI were used to visualize and score aggregates. The number of scored nuclei is indicated (n). **$p$-value <0.01, using the $\chi_2$ test. (D, E) Quantification of PABPN1 nuclear aggregate cross-sectional areas. Each aggregate was delimited in a focal plan and the surface was calculated using ImageJ. (D) Mean values of areas with SD are indicated in arbitrary units. The number of scored aggregates is indicated (n). (E) The distribution of cross-sectional areas is presented as box plots. The boxes represent 50% of the values; the horizontal lines within the boxes correspond to medians. *$p$-value <0.05, using the unpaired Student's $t$-test. Quantifications in C-E were from two independent experiments. (F) Western blot of thoracic extracts revealed with anti-PABPN1 showing that the total amounts of PABPN1 are not affected by feeding with MG132. α-Tubulin was used as a loading control.

## Discussion

In this paper, we functionally address the role of the UPS in OPMD pathogenesis using *Drosophila* OPMD models. Genetic screens have identified a large number of UPS components involved in the different steps of UPS-dependent protein degradation, including ubiquitin conjugating-enzymes, ubiquitin ligases and proteasome subunits, as contributing to OPMD.

The genetic data reveal that decreasing the gene dosage of UPS components reduce OPMD muscle defects, suggesting that elevated UPS activity play an important role in OPMD pathogenesis. Molecularly, we find that the rescue of muscle defects in the presence of mutants for proteasome subunits can be uncoupled from reduced PABPN1-17ala aggregation. In contrast, the presence of these mutants consistently decreases the degradation of myofibrillar proteins. These data reveal that deregulation of the UPS in OPMD does not contribute to pathogenesis through PABPN1 aggregation, but through activated degradation of muscle proteins.

Proteasome dysfunction has been described to be a major contributor of pathogenesis in several neurodegenerative diseases that involve protein aggregation, leading to the actual accumulation of aggregates [1,59]. Proteasome activity is known to decline with age, and this decline might participate in accumulation of pathological misfolded proteins. In addition, a common mechanism of proteasome inhibition has been reported to depend on small misfolded protein oligomers that stabilizes the proteasome in its closed gate conformation, thus preventing its activity [59,60]. Although some level of proteasome inhibition by a similar mechanism involving PABPN1 oligomers is likely in OPMD, our functional data indicate that this inhibition is counteracted by other molecular events (e.g. enhanced expression of UPS genes) that result in increased UPS activity towards myofibrillar proteins and eventually muscle atrophy (see Fig 8 for a model). Indeed, we found that myofibrillar organization and sarcomeric structure were progressively lost in OPMD muscles, and that these defects along with the loss of sarcomeric proteins were reduced in the presence UPS heterozygous mutants.

The loss of muscle mass or atrophy also strongly depends on protein degradation by the UPS and a key actor of UPS-mediated atrophy is the E3 ubiquitin ligase MuRF1/TRIM63 that is directly involved in the degradation of myofibrillar proteins [61,62]. Molecular analyses of a mouse model of OPMD in which PABPN1-17ala is specifically expressed in muscles, have shown that atrophy plays a key role in OPMD pathogenesis [63]. Transcriptomic analyses of OPMD mouse muscles at three time points have identified muscle atrophy as one of the strongly deregulated pathways, one third of the genes progressively deregulated with time being associated with atrophy. These data were confirmed by histological and functional studies that also revealed progressive atrophy quantified through the reduction of muscle mass and associated with a reduction of the muscle strength. In addition, atrophy was confirmed in pharyngeal muscles of OPMD patients [64]. Importantly, in this mouse OPMD model, muscle atrophy correlates with increased levels of *MuRF1/TRIM63* mRNA and increased proteasomal activity, thus suggesting that enhanced UPS-dependent muscle protein degradation does contribute to OPMD pathogenesis [63]. Furthermore, this notion is reinforced by studies of the *Pabpn1-17ala/Pabpn1* OPMD mouse model that reproduces the disease genotype of OPMD patients [24]. These mice show a mild myopathic phenotype and reproduce the early mitochondrial defects, previously identified in the *Drosophila* model and confirmed in OPMD patients [23]. Strikingly, a proteomic analysis of the rectus femoris muscle from this mouse model revealed that, whereas depleted proteins were enriched in mitochondrial proteins, more abundant proteins were most significantly enriched for terms related to the proteasome complex ("proteasome", "proteasome accessory complex") [24]. The capacity of UPS mutants to reduce OPMD muscle defects in *Drosophila* demonstrates the functional importance in OPMD pathogenesis of increased UPS activity that leads to enhanced degradation of myofibrillar proteins. It should be noted that an homologue of *MuRF1/TRIM63* is not present in the *Drosophila* genome [62]; therefore, the contribution of this E3 ubiquin ligase in OPMD could not be analyzed in *Drosophila*.

Intriguingly, increased UPS activity might be a common mechanism leading to atrophy in various myopathies. An increased proteasome activity together with higher levels of *Atrogin-1* mRNA that encodes a muscle specific E3 ubiquitin ligase were reported to contribute to

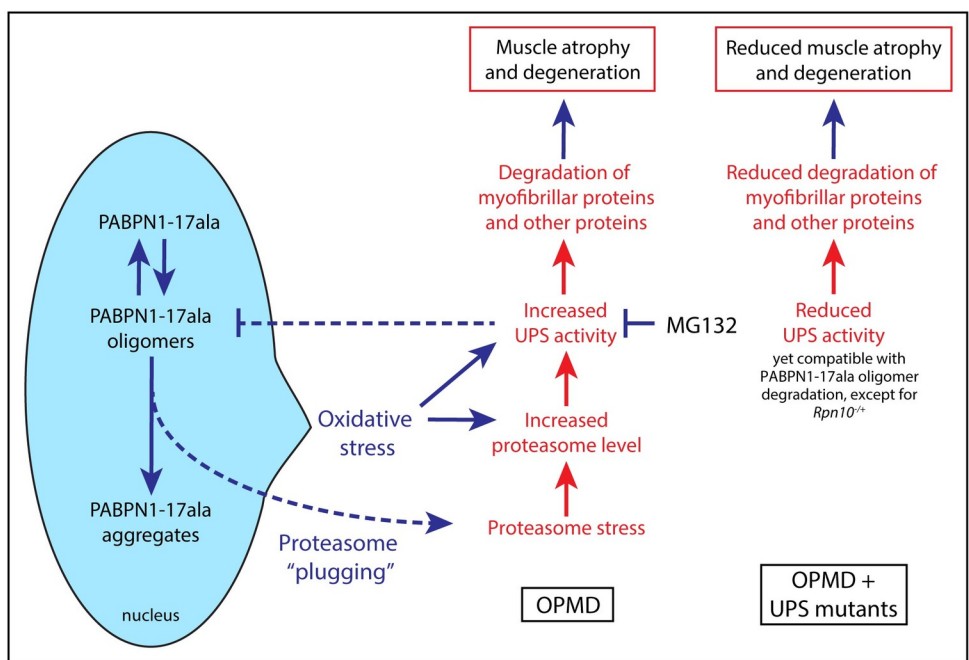

**Fig 8. Model of the UPS involvement in OPMD pathogenesis.** PABPN1-17ala is present under three forms in OPMD muscles: soluble monomers, oligomers, and aggregates produced from oligomers. Oligomers can be degraded by the 26S proteasome; however, they tend to inhibit it by plugging its channel. PABPN1-17ala expression triggers oxidative stress by reducing mitochondrial activity. This, together with proteasome plugging with PABPN1-17ala oligomers lead to increased proteasome levels through transcriptional activation. Other UPS components important for myofibrillar protein degradation would also be activated, leading with higher proteasome levels to enhanced degradation of myofibrillar proteins and pathology. In the presence of heteozygous mutants of proteasome subunits, proteasome levels might increase through transcriptional compensatory regulation [81]. However, proteasomes are less active than normal, leading to decreased degradation of myofibrillar proteins. Nevertheless, the gain in overall proteasome activity might allow for more degradation of PABPN1-17ala oligomers and thus lower aggregation. In the presence of *Rpn10* heterozygous mutant, proteasome assembly would be affected, leadind to decreased overall activity. Consequently both myofibrillar protein and oligomer degradation would be reduced, resulting in more myofibrillar proteins and enhanced PABPN1-17ala aggregation. Similarly, oral treatment with MG132 reducing the overall proteasome degradative capacity, would lead to improvement of myofibrillar protein levels, together with decreased degradation of PABPN1-17ala oligomers and thus more aggregates.

muscle weakness and atrophy in a mouse model of myotonic dystrophy type 1 [65]. In this model, increased expression of UPS genes was recorded at an early time point preceding the onset of muscle weakness, consistent with the implication of higher proteasome activity in the pathogenesis [65]. Such an early UPS gene upregulation was also reported in the *Pabpn1-17ala/Pabpn1* OPMD mouse model [24].

The molecular basis of UPS gene upregulation in OPMD is currently unknown. A direct role of PABPN1 in alternative polyadenylation of UPS genes has been proposed, and a change in poly(A) site usage was shown to correlate with increased levels of *Atrogin-1* mRNA [66]. However, the mouse and cell models used in this study involved solely a downregulation of *Pabpn1* that, although likely contributing, does not completely explain OPMD pathology [24]. Alternative polyadenylation was not globally affected in the *Pabpn1-17ala/Pabpn1* mouse model pointing to the implication of other molecular mechanisms underlying UPS gene upregulation in OPMD. Regarding the proteasome, its level is known to increase upon proteotoxic stress through enhanced transcription. In mammals, this enhancement is mediated by the transcription factors Nrf1 -activated upon proteasome impairment- and Nrf2 -activated upon oxidative stress- [67,68], and this function of Nrf2 is conserved in *Drosophila* [69,70].

Interestingly, oxidative stress through mitochondrial dysfunction is one of the earliest defects in OPMD [23]. In addition, reactive oxygen species were shown to enhance proteasome activity in a *Drosophila* model of amyotrophic lateral sclerosis (i.e. mutant of the gene encoding superoxide dismutase 1) [71]. Therefore, it is likely that oxidative stress contributes to proteasome upregulation in OPMD. Furthermore, PABPN1 oligomers might also participate in this process through proteasome inhibition [59,60] and subsequent activation of the proteasome stress response (Fig 8). This sequence of events might participate in the progressivity of the disease as proteasome upregulation would be the consequence of defects arising earlier. Thus, we propose the following molecular model contributing to OPMD progression. Reduced mitochondrial activity due to defective polyadenylation and poly(A) tail shortening of mRNAs encoding subunits of the respiratory chain arises early in OPMD, leading to oxidative stress [23,24]. This, together with accumulation of PABPN1 oligomers that might slowly build up in patients would induce proteasome upregulation via transcriptional activation. Higher levels of proteasome and other UPS components would lead to increased ubiquitination and degradation of myofibrillar proteins, resulting in turn in muscular atrophy (Fig 8).

An important information from our study is the proof-of-concept that drug treatment decreasing proteasome activity is beneficial for OPMD in an animal model. We find that oral treatment with MG132 significantly reduces OPMD muscle defects quantified through the number of flies showing affected wing position. This improvement of muscle function is not linked to reduced PABPN1-17ala aggregation, since aggregation is enhanced following MG132 treatment. The proteasome was reported to regulate PABPN1 stability, therefore, this increase in PABPN1-17ala aggregation after MG132 treatment was expected. These data reinforce the notion that aggregates *per se* are not likely to be the major trigger of OPMD pathogenesis. In addition, aggregates might also be protective by titrating misfolded PABPN1 and/ or toxic oligomers, as it has been shown for other proteinopathies [72]. Other conditions have been reported where improvement of muscle function in OPMD animal models is uncoupled from the decrease of PABPN1 aggregation [23]. In particular, acting directly to increase muscle mass through myostatin inhibition in a mouse model of OPMD, improves muscle function without decreasing PABPN1 aggregation [73]. A number of proteasome inhibitors are approved drugs that are routinely used in clinic for the treatment of myeloma and lymphoma [74]. Our data provide the first evidence that pharmacological inhibition of proteasome activity might represent a new therapeutic approach for OPMD.

## Materials and methods

### *Drosophila* stocks and genetics

$w^{1118}$ was used as control. Flies were raised at 25˚C unless specified otherwise. The stock $w^{1118}$; *UAS-PABPN1-17ala; 24B-Gal4/ TM6B, tubP-Gal80, Tb$^1$* was generated to perform the screens. Deficiency (Df) and mutant stocks were obtained from the Bloomington *Drosophila* Stock Center and the Vienna *Drosophila* Resource Center. The lists of Df and mutant alleles are provided in S1 and S2 Tables. $w^{1118}$; *UAS-PABPN1-17ala; 24B-Gal4/ TM6B, tubP-Gal80, Tb$^1$* females were crossed to *Df/Balancer* or *mutant/Balancer* males at 22˚C. For each experiment, control crosses were performed alongside experimental crosses with females of the same genotype and either $w^{1118}$, Canton-S or $w^{1118}$; *UAS-LacZ* males for negative controls, and $w^{1118}$; *UAS-3F5* males for positive controls. 3F5 is an anti-PABPN1 nanobody whose expression in muscles reduces OPMD muscle defects [31]. Five days after mating, males and females were discarded and crosses were kept at the same temperature until pupa formation. Pupae were scored in the tube and the percentage of [Tb$^+$] pupae was determined. Crosses with negative control males produced 5–15% [Tb$^+$] pupae. Experimental crosses producing >30% [Tb$^+$]

pupae identified suppressive Df. The suppressive regions within large Df were identified by setting up similar crosses with smaller Df spanning the large suppressive Df. These smaller Df were choosen using cytologically or molecularly characterized Df breakpoints, as well as the CytoSearch tool from FlyBase. The suppressor genes within suppressive regions were identified by setting up again similar crosses with mutant alleles, or exceptionally RNAi, of candidate genes. Candidate genes were determined using literature as well as Endeavour-HighFly, a software for computational prioritization of candidates genes [42]. Suppressor genes were identified by the presence of >30% [Tb$^+$] pupae in the cross progeny. Mutant alleles of UPS components were analyzed using the same approach. The stock $w^{1118}$; *Act88F-PABPN1-17ala*, *Mhc-Gal4* was used to express *Pomp* RNAi in OPMD muscles. The *Pomp$^{GLC01750}$* stock carries *Pomp* RNAi at the *attP2* site (*P{y[+t7.7] v[+t1.8] = TRiP.GLC01750}attP2*). The stock containing the landing site *attP2* alone (*P{caryP}attP2*) was used as a control.

## Preparation of drug supplemented medium and analysis of drug toxicity

Drug supplemented food was prepared as follows. Instant *Drosophila* medium (Carolina Biological Supply Company) was reconstituted in each vial with a solution of 1% yeast in water, supplemented with either increasing concentrations of MG132 (Santa Cruz Biotechnology sc-201270) diluted in DMSO (dimethyl sulphoxide, Sigma D2650), or DMSO alone. Each vial contained 5 ml or 2.5 ml of reconstituted medium, to raise larvae or adults, respectively. Individuals were fed with the drug from larval stages as follows: 70 to 80 first instar larvae were transferred per vial and developed in the same vial up to adulthood. 20 adults were transferred per vial in new vials with the same concentration of drug. Adults were transferred every three days to new vials containing fresh drug-supplemented food. To determine drug toxicity, 80 first instar larvae were transferred into vials containing drug- or DMSO alone-supplemented medium. Individuals reaching adulthood were scored.

## Analysis of wing posture and adult musculature

Analysis of abnormal wing posture and visualization of thorax muscles under polarized light were performed as described previously [40]. Muscles were scored from two to three independent crosses.

## Immunostaining and visualization of muscle structure

Embryonic and larval musculature was revealed using staining with phalloidin-ATTO 633 (Sigma 68825) 1:500. Staged embryos and dissected larval muscles were prepared according to standard procedure [75]. Dissection and immunostaining of adult fly hemi-thoraxes to analyze sarcomeric structure were performed as previously described [76]. The following primary antibodies were used: mouse anti-Mhc (DHSB, 3E8-3D3) 1:50; rat anti-Kettin (Abcam, ab50585) 1:500. Immunostaining of adult IFMs to quantify aggregates were performed as previously reported [40]. Dilutions of primary antibodies were as follows: rabbit anti-PABPN1 1:1000; rat anti-p62 (a gift from S. Gaumer) 1:500; mouse mono- and polyubiquitinated conjugates monoclonal antibody (FK2, Enzo Life Sciences) 1:500. DNA was labeled with 1 µg/ml DAPI (Sigma-Aldrich). Confocal sections were acquired using a Leica SP8 confocal scanning microscope. PABPN1 nuclear aggregates were identified based on anti-PABPN1 staining and the lack of DAPI staining; their surfaces were measured using ImageJ. Quantification of p62 aggregates was performed as follows. Images were analyzed using the Analyze Particules tool in ImageJ. The parameters for particule surface and circularity were set up to identify three classes of aggregates: Class I, diameter >0.25 and <1.5 µm; Class II, diameter 1.5–1.9 µm; Class III diameter >1.9 µm; circularity range allows identification of elongated polygon to perfect circle.

## Western blots

Western blots of adult indirect flight muscles were performed as reported previously [40]. For all western blots, the protein used as a loading control was revealed on the same western blot. Regular protein samples were prepared by direct grinding of dissected thoraxes in Laemmli buffer, followed by 5 min incubation at 95˚C. Myofibrillar proteins were extracted from dissected thoracic muscles as follows. Muscles from ten thoraxes were dissected in 300 μL of lysis buffer (5 mM Tris pH 7.5, 5 mM EDTA, 1 mM PMSF, protease inhibitor cocktail (10 μL/mL of lysis buffer), 10 mM NEM, 1% Triton X100) [77] and centrifuged at 10,000 g and 4˚C for 10 min. After removal of the supernatant containing the cytoplasmic fraction, the pellet enriched in myofibrillar proteins was resuspended in 100 μL of lysis buffer and sonicated with five cycles of 30 sec ON/30 sec OFF sonication using the Bioruptor Pico. Myofibrillar protein extracts were conserved at -80˚C before loading with Laemmli buffer on acrylamide gels. Antibody dilutions for western blots were as follows: rabbit anti-PABPN1 [78] 1:2500; mouse anti-Ubiquitin Ubi-1 (13–1600, Life Technology) 1:1000; rat anti-myosin (MAC 147, Abcam ab51098) 1:10; mouse anti-α-tubulin (T5168, Sigma-Aldrich) 1:5000; rabbit anti-RpL10Ab [79] 1:5000; rabbit anti-Rpn11 [80] 1:300; rabbit anti-Prosα2 [80] 1:500; mouse anti-Prosα7 (PW8110, Enzo Life Sciences) 1:1000.

## Proteasome activity *in vitro* assays

Proteasome activity was analyzed using the fluorescent substrate N-succinyl-leucine-leucine-valine-tyrosine-7-amino-4-methylcoumarin (Suc-LLVY-AMC, I-1395, Bachem), specific for the chymotrypsin-like peptidase activity of the proteasomes. The samples were prepared from 10 thoraxes dissected on dry ice and lysed in 120 μL of ice-cold buffer (50 mM Tris-HCl pH 8.0, 0.5% IGEPAL, 5 mM $MgCl_2$, 1 mM ATP, 1 mM DTT, 0.5 mM EDTA, 10% glycerol and complete protease inhibitor mixture (11697498001, Roche)). The lysis was achieved by three cycles of grinding with a pestle during 1 min on ice followed by 5 min homogenization at 4˚C on a rotator. Lysates were centrifuged at 16,000 g for 20 min at 4˚C to remove debris, and the supernatants were transferred to new tubes. Protein concentration was measured using Bradford assay and BSA as a reference. Chymotrypsin-like protease activity was measured as follows. 50 μg of protein extracts were incubated in 50 μL of assay buffer (20 mM Tris-HCl pH7.5, 5 mM $MgCl_2$, 10% glycerol, 0.5 mM EDTA, 1 mM ATP, 1 mM DTT, 0.2 mM Suc-LLVY-AMC) in a black flat-bottom 96-well plate (Greiner bio-one). Fluorescence of the released 7-amido-4-methylcoumarin (AMC) was measured at 380 nm excitation wavelength and 440 nm emission wavelength, every 5 min for 60 min at 37˚C using a Fluostar Optima 96-well plate reader. The reaction was quenched by the addition of sodium dodecyl sulfate (SDS) to a final concentration of 1%. Fluorescence was recorded as arbitrary units and measurements at 20 min are shown. Measurements were from three biological replicates quantified in triplicate. For the zymogram of chymotrypsin-like proteasome activity in native gels, 15 μg of protein extracts were loaded onto 3.8% acrylamide native gels. Electrophoresis was performed for 3 to 4 h at 150 V in migration buffer (90 mM Tris-base, 90 mM boric acid, 0.1 mM EDTA, 0.5 mM ATP, 5 mM $MgCl_2$, 0.5 mM DTT). The chymotrypsin-like proteasome activity was visualized by incubating gels for 20 min at 37˚C in 100 μM Suc-LLVY-AMC in migration buffer. The fluorescence was visualized using an UV imaging system.

## RT-qPCR

RT-qPCR were performed as previously described [23]. The primers were as follows.
   *Rpn10*: TCAGTGGCAAAGTGACAAGC; GTTGGAAAGTAGTCTCCGTTG
   *Rpn11*: TGAGCACTGTTCCATCAACG; TCGGGGCGTCATTTTCTCCTC

*Prosβ4*: CGCCCGATCCATTATAGTGAT; TTGCGCATCTTGTACAACGC
*Pomp*: CCCTCAAGATGGGCATGGAG; TGTTCTCGGGCAAGTTCATG
*mib2*: CAAATTGAGCCAACCAAACGC; AACGTCCGATTTCGCAAACG
*sop*: CACCCCAATAAAGTTGATAGACCT; ACCACCACGAGAGCCAAAT

## Statistical analyses

All statistical analyses were performed using GraphPad Prism software (GraphPad, La Jolla, CA, USA).

## Supporting information

**S1 Table. Results of the genome-wide screen.**
(XLSX)

**S2 Table. List of UPS genes tested in the second screen and results of the screen.**
(XLSX)

**S3 Table. Numerical values of data in Figs 4, 5 and 7.**
(XLSX)

**S1 Fig. Charaterization of the *PABPN1-17ala/+; 24B-Gal4/+* OPMD model.** (A) Musculature of late *PABPN1-17ala/+; 24B-Gal4/+* and control *24B-Gal4/+* embryos visualized using phalloidin staining. No major muscle defects were visible (n = 46 and 52 *PABPN1-17ala/+; 24B-Gal4/+* and control embryos, respectively). Scale bar: 10 μm. (B) Musculature of *PABPN1-17ala/+; 24B-Gal4/+* and control *24B-Gal4/+* third instar larvae visualized using phalloidin staining. Three adjacent hemi-segments are shown. Arrows point to very thin muscles, arrowheads to splitted muscles, and stars to broken muscles. All muscles in OPMD larvae are thiner than in control larvae. Scale bar: 100 μm. (C) Quantification of affected muscles visualized in B. The number of hemi-segments with defective muscle fibers described in B were scored. Note that in *24B-Gal4/+* control larvae, a single muscle fiber was defective per affected hemi-segment. ****$p$-value <0.0001 using the $\chi_2$ test.
(PDF)

**S2 Fig. Number and location of suppressive deficiencies identified in the genome-wide screen.** (A) Number of positive deficiencies from the Deficiency kit, identified in the genome-wide OPMD screen, per chromosome arm. The Deficiency kit from the Bloomington *Drosophila* Stock Center corresponds to a set of large deficiencies covering most of the *Drosophila* genome. (B) Localization of suppressive regions identified in the genome-wide OPMD screen, per chromosome arm. Suppressive regions were identified using large deficiencies from the Deficiency kit and smaller deficiencies spanning the large positive deficiencies.
(PDF)

**S3 Fig. Characterization of UPS mutants.** (A) Schematic representation of UPS gene mutations used from Fig 3. The depicted mutations correspond to *P*-element or *piggyBac* insertions, except for *mib2[1]*. Thin lines indicate intergenic regions and introns, open boxes represent UTRs and black boxes represent coding sequences. (B) Quantification of mRNA levels in the corresponding heterozygous mutant compared to wild type using RT-qPCR. RNAs were prepared from thoraxes of the indicated genotypes at day 6. *sop* was used as a control mRNA. Quantification in triplicates of three biological replicates. *$p$-value <0.05, ns: non significant, using the unpaired Student's *t*-test. (C) Decrease of wing position defects following expression of *Pomp* RNAi in muscles with *Mhc-Gal4*. Left panel: Percentages of flies with abnormal wing position were scored at day 6; the numbers of scored flies are indicated (n). *attP2* is the

insertion site in which *Pomp*-RNAi was introduced and the stock bearing *attP2* alone was used as control. **** *p*-value <0.0001, using the $\chi_2$ test. Right panel: quantification of *Pomp* mRNA levels using RT-qPCR. Legend is as in B. ** *p*-value <0.01, using the unpaired Student's *t*-test. (D) Effect of *Prosβ1* heterozygous mutant in OPMD adult flies. Percentages of flies with abnormal wing position were scored at day 6; the numbers of scored flies are indicated (n). **** *p*-value <0.0001, using the $\chi_2$ test. Abbreviations and genotypes used in C and D are indicated. (PDF)

**S4 Fig. Decrease of muscle degeneration with mutants of UPS components.** (A) IFMs in OPMD flies in the presence of UPS heterozygous mutants visualized under polarized light at day 11. IFMs are less affected in all genotypes compared to OPMD IFMs alone. Quantification of affected muscles is shown in Fig 3D. (B) Western blots of thoracic extracts revealed with anti-PABPN1 showing that the total amount of PABPN1 is not affected by mutants of the UPS components. α-Tubulin was used as a loading control. (PDF)

**S5 Fig. Quantification of proteasome subunits in wild-type and OPMD muscles and of proteasome chymotrypsin-like activity in wild type and UPS heterozygous mutants.** (A) Western blots of protein extracts from thoracic muscles of wild-type and OPMD (*Act88F-PABPN1-17ala/+*) flies at days 3 and 6 revealed with antibodies against three proteasome subunits: Prosα2, Prosα7 and Rpn11. α-Tubulin was used as a loading control. Quantification was performed using the ImageJ sofware with three to four biological replicates. Error bars represent SD. **** *p*-value <0.0001, *** *p*-value <0.001, ** *p*-value <0.01, * *p*-value <0.05, using the unpaired Student's *t*-test. (B) Proteasome chymotrypsin-like activity of protein extracts from wild-type and heterozygous UPS mutant thoracic muscles, at day 3 and day 6 of adulthood. Chymotrypsin-like activity was quantified by measuring AMC fluorescence following hydrolysis of Suc-LLVY-AMC. Means of three biological replicates quantified three times. Error bars represent SEM. **** *p*-value <0.0001, * *p*-value <0.05, ns: non significant, using the unpaired Student's *t*-test. (PDF)

**S6 Fig. Weak defects in OPMD (*Act88F-PABPN1-17ala/+*) thoracic muscles at day 3.** (A) IFMs in wild-type and *Act88F-PABPN1-17ala/+* thoraxes visualized under polarized light at day 3. Six DLMs and seven DVMs were scored per hemi-thorax. The white arrow indicates a slight defect in a DLM, which is representative of the weak defects visible at that time. (B) Quantification of affected muscles. The numbers of scored DLMs or DVMs are indicated (n). No defects were observed in indirect flight muscles of wild-type flies (n = 54 DLMs and 105 DVMs). (PDF)

## Acknowledgments

We are grateful to S. Dorner, S. Gaumer, M. Miura and S. Murata for their gifts of antibodies. We thank Bloomington *Drosophila* Stock Center for keeping and sending *Drosophila* stocks. We thank Julie Cremaschi for analyzing proteomic data from the *Pabpn1-17ala/Pabpn1* mouse model.

## Author Contributions

**Conceptualization:** Martine Simonelig.

**Data curation:** Aymeric Chartier, Sandy Al Hayek, Rima Naït-Saïdi.

**Formal analysis:** Cécile Ribot, Cédric Soler, Aymeric Chartier, Nicolas Barbezier, Olivier Coux.

**Funding acquisition:** Martine Simonelig.

**Supervision:** Olivier Coux, Martine Simonelig.

**Writing – original draft:** Cécile Ribot, Martine Simonelig.

**Writing – review & editing:** Martine Simonelig.

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
