## [Decision Letter · Decision Letter 0]

24 May 2021

Dear Dr Simonelig,

Thank you very much for submitting your Research Article entitled 'Activation of the ubiquitin-proteasome system contributes to oculopharyngeal muscular dystrophy through muscle atrophy' to PLOS Genetics.

The manuscript was fully evaluated at the editorial level and by independent peer reviewers. The reviewers appreciated the attention to an important problem, but raised some substantial concerns about the current manuscript. Based on the reviews, we will not be able to accept this version of the manuscript, but we would be willing to review a much-revised version. We cannot, of course, promise publication at that time.

If you decide to revise the manuscript for further consideration at PLOS Genetics, please aim to resubmit within the next 60 days, unless it will take extra time to address the concerns of the reviewers, in which case we would appreciate an expected resubmission date by email to plosgenetics@plos.org.

[LINK]

We are sorry that we cannot be more positive about your manuscript at this stage. Please do not hesitate to contact us if you have any concerns or questions.

Yours sincerely,

Bingwei Lu

Associate Editor

PLOS Genetics

Gregory Barsh

Editor-in-Chief

PLOS Genetics

Reviewer's Responses to Questions

**Comments to the Authors:**

Reviewer #1: In this manuscript, Ribot et al. examine the role that increased proteasome activity plays in a drosophila model of oculopharyngeal muscular dystrophy, which is due to polyA tract expansion in the PABPN1 nuclear protein. The authors designed a screen based on genomic deficiencies to identify modifiers of developmental lethality induced by expression of PABPN1 in skeletal muscle. They identify components of the ubiquitin proteasome as responsible for muscle degeneration induced by PABPN1. The authors went on to test the impact of heterozygous mutations for some components of the proteasome and associated proteins and found that they rescue muscle degeneration (as estimated with histological analyses and assessment of wing positioning). They also test whether MG132, a proteasome inhibitor, can reduce PABPN1-induced muscle degeneration and found it to be the case. The authors also report that preservation of muscle integrity occurs independently from PABPN1 aggregates, and suggest that proteasome hyperactivation (induced by mutant PABPN1) is the key reason for muscle degeneration.

Overall, this manuscript provides a sound take-home message given that while the proteasome is needed to degrade aggregation-prone proteins and for normal protein turnover, it is also known to cause muscle atrophy when overactive. Therefore it is not surprising that preventing its overt activation may be protective for muscle mass in certain conditions such as oculopharyngeal muscular dystrophy.

This reasonable take-home message is however not completely supported by the data included in this manuscript. There are a number of inconsistencies that weaken the manuscript and some assays do not appear robust, as explained in detail here below. In particular, most of the figures in the manuscript (Figure 3 onward) are based on the use of a series of heterozygous mutants for proteasome components (Rpn10, Rpn11, ProsB4, Pomp) and for an E3 ligase (mib) which are used as tools to inhibit the proteasome. The authors find that in all cases these heterozygous mutants can rescue PABN1-induced muscle degeneration (Figure 3,5,6). However, whereas Rpn10 heterozygous mutants reduce proteasome activity (Fig. 4A), heterozygous mutants of Rpn11, ProsB4, Pomp, and mib do not affect the proteolytic activity of the proteasome (Fig. 4). On this basis, there is currently little evidence to conclude that proteasome inhibition is responsible for reducing muscle degeneration induced by PABPN1 as only Rpn10 heterozygous mutants seem to impact proteasome function. Moreover, it remains unknown how heterozygous mutations of Rpn11, ProsB4, Pomp, and mib2 rescue muscle degeneration via mechanisms unrelated from proteasome function.

Specific comments:

Figures 1-2: these figures report schemes that summarize the results of the screen. It would be better to provide a summary with quantitative data, particularly for selected proteasome genes shown in Figure 2.

Figure 2: It is surprising to see several E3 ligases that rescue PABPN1-induced degeneration. E3 have normally specific set of target proteins and therefore normally have unique biological functions. Likewise, because DUBs can oppose E3 function by removing polyubiquitin chains, it is surprising to see that they score like E3s. It would be helpful if the author could show the quantitative data and provide information on the controls used in these experiments.

Figures 3-4: The authors use a series of heterozygous mutants for proteasome components (Rpn10, Rpn11, ProsB4, Pomp) and an E3 ligase (mib) to test whether they can be used to inhibit the proteasome. They find that in all cases these heterozygous mutants can rescue PABN1-induced muscle degeneration (Figure 3).

However, whereas Rpn10 heterozygous mutants reduce proteasome activity (Fig. 4A), heterozygous mutations of Rpn11, ProsB4, Pomp, and mib do not affect the proteolytic activity of the proteasome (Fig. 4). On this basis, how do the authors explain the rescue of muscle degeneration by heterozygous mutations of Rpn11, ProsB4, Pomp, and mib2? There is currently little basis to conclude that proteasome inhibition is responsible for reducing muscle degeneration induced by PABPN1 as only Rpn10 heterozygous mutants seem to impact proteasome function.

Moreover, if not by impacting the proteasome, how would heterozygous mutations of Rpn11, ProsB4, Pomp, and mib rescue PABPN1-induced muscle degeneration?

The heterozygous mutants for proteasome components here used have not been characterized molecularly. What is the evidence that they reduce the mRNA or protein levels of the targeted genes (Rpn10, Rpn11, ProsB4, Pomp, and mib2)? qRT-PCR analysis should be sufficient to address this point.

The authors use whole-body heterozygous mutants for proteasome components. The conclusions would be strengthened by using transgenic modulation (such as RNAi) of these or other proteasome components targeted to skeletal muscle.

Experiments with isogenic genetic background would be useful to avoid confounding effects deriving from genetic background mutations. At present, it seems that all experiments have been done with varied genetic backgrounds which could impact the outcome of these experiments.

Figure 5: In this figure, the authors have scored the number of nuclei with PABPN1aggregates. They report that PABPN1 aggregates slightly increase in response to proteasome inhibition and on this basis argue that the aggregates per se are not the cause of muscle atrophy. While this reviewer agrees with this notion, the supporting data is weak.

Specifically, changes in the number of nuclei with aggregates as well as in the nuclear aggregate cross-sectional area appear minor, even if statistical significant (which is due to the high number of nuclei scored).

There is also some inconsistency across interventions: in Fig. 5D the nuclear aggregate cross-sectional area increases for Rpn10-/+, does not change for Prosb4-/+, and decreases for Pomp+/-. As these mutations are all supposed to affect the proteasome, it is puzzling on why they would score differently. But most importantly, why do the authors continue to use all these mutants? From studies in Fig. 4, they know that only Rpn10 mutants affect the proteolytic activity of the proteasome so I do not see any basis to employ other heterozygous mutants to inhibit the proteasome if in fact they do not.

There is inconsistencies in the impact of these proteasome mutations on “the percentage of nuclei with aggregates“ (Fig. 5B) versus the “nuclear aggregate cross-sectional area” (Fig. 5D). One would expect these two estimates to be largely overlapping but they are not.

Based on the results in Fig. 4 that have shown an effect on proteasome activity only for Rpn10 mutations, I would have expected only Rpn10 mutations to score and that other mutants would have no effect. Instead Fig. 5B reports no effect of Rpn10-/+ on the percentage of nuclei with aggregates.

The authors should provide an explanation for these inconsistencies and use more robust (biochemical) methods to detect aggregate versus oligomeric PABPN1 if they want to draw conclusions on how inhibiting the proteasome affects PABPN1 aggregation.

Comment: PABPN1 aggregation is described as a deleterious aspect. While that is certainly a disease biomarker that indicates loss of function of PABPN1, it has been shown in other contexts that it is actually protective that aggregation-prone proteins are sequestered in defined compartments (aggregates or aggresomes), in order to reduce their interaction with native proteins. There are indeed several studies in other diseases showing that oligomers can be more toxic than larger aggregates.

Figure 6: same issues as above. The authors use heterozygous mutants for proteasome components that do not impact proteasome function (as demonstrated in Fig. 4) but they draw conclusions from these experiments as if these mutants would be inhibiting proteasome function.

Figure 7: The authors show that feeding larvae with a proteasomal inhibitor (up to 600uM of MG132) reduces wing positioning defects due to mutant PABPN1. Why is this drug provided from larval development, what is the rationale for this? In fact, because there is no feeding through pupal development and because most larval muscle are histolyzed during pupal development, it is unlikely that larval feeding of MG132 has a direct impact in preserving myofibrils as these are degraded anyway during the pupal stages of development, apart for persistent muscles. Moreover the PABPN1 transgene seems to be expressed under the control of Act88F, which is not expressed in larval muscles. On this basis, it is unclear how feeding MG132 at the larval stages will impact aggregation and toxicity of PABPN1 in adult muscles at day 1. What is the reason of this larval feeding regimen? Are the same results obtained with MG132 feeding restricted to adult flies?

Reviewer #2: The authors use a Drosophila model of oculopharyngeal muscular dystrophy (OPMD), made by expression of a mutant mammalian poly(A) binding protein nuclear 1 (PABPN1) protein containing an expanded polyalanine tract, to follow up on previous transcriptomic work showing a deregulated ubiquitin-proteasome system (UPS). OPMD is a complex muscle disease, wherein multiple functions are affected due to the presence of the mutant protein. Here the authors focus on protein degradation and PABPN1 aggregation. They used both targeted and genome-wide genetic screening for improved muscle function to identify specific UPS components whose downregulation reduces myofibrillar protein degradation. These do not consistently reduce PABPN1 aggregation. This contrasts with previous work showing that reducing PABPN1 aggregation improves muscle function. Further, inhibition of the proteasome yields similar results, suggesting a potential therapeutic approach for disease treatment. This is a well-done paper containing a tremendous amount of experimental data, with a strong genetic component. The following comments should be considered by the authors:

1. It would be helpful to provide more rationale for the Drosophila model. Why is the mammalian mutant PABPN used instead of a mutant Drosophila protein? Where is the mutant gene expressed and by what methodology? This may have been established in previous papers, but is important enough an issue to mention here. Further, it would be useful to point out that while it is a hybrid model, the screen may prove valuable for therapeutic use in that it is actually testing how defects affiliated with the mammalian protein can be suppressed.

2. While the genome-wide screen seems extremely effective in identifying suppressors, the fact that the suppressors were in previously identified pathways is not particularly surprising. This is because the investigators tested candidate genes (at least partially) based upon those with reported relationships with OPMD and/or PABPN1. This should be pointed out in the paper.

3. The finding that reduction in expression of 77% of tested UPS genes suppressed lethality was impressive. Do the authors care to speculate why the remaining 23% did not? In this regard, the IFM defect suppression experiment is quite convincing, but did the authors try a negative control, i.e., a mutant for one of the UPS components that did not suppress lethality in the directed screen?

4. For the western blots in Fig. 3E and Fig S2, how were the samples prepared? Presumably not by the myofibrillar protein isolation procedure given in the Methods section. For all western blots: are the blots for the control protein from the same gel or at least the same samples? This should be stated.

5. Please speculate in the text on how the mutants work to reduce proteasome function, if activity of the proteasome is maintained in most of them (Fig. 4)?

6. It would be better to state in the abstracts and elsewhere that the improved muscle structure/function observed due to reducing UPS components does not correlate with the levels of aggregate accumulation, since some of the current wording implies there is no change in aggregate accumulation.

7. Why not include the statistical significance indicators in Fig. 5C?

8. Why was tubulin replaced with a different control protein in Fig. 6E?

9. It would be helpful to show (or refer to from another publication) a confocal or TEM image of sarcomeres from PABPN1 muscles and from PABPN1 muscles that have been rescued. Is there sarcomere degradation that is prevented?

10. The authors might mention that aggregates could serve a useful function in removing misfolded proteins from the soluble cytoplasm to prevent their aberrant interactions.

11. A number of the statistical comparisons (e.g., Fig 7C-E) compare only two independent experiments. Does PLoS permit this?

Reviewer #3: see attachment

**Have all data underlying the figures and results presented in the manuscript been provided?**

Reviewer #1: Yes

Reviewer #2: **No: **Numerical data are not provided for a few figure panels, just summary statistics and plots.

Reviewer #3: Yes

PLOS authors have the option to publish the peer review history of their article (what does this mean?). If published, this will include your full peer review and any attached files.

Reviewer #1: No

Reviewer #2: **Yes: **Sanford I. Bernstein

Reviewer #3: No

---

## [Decision Letter · Decision Letter 1]

14 Dec 2021

Dear Dr Simonelig,

Thank you very much for submitting your Research Article entitled 'Activation of the ubiquitin-proteasome system contributes to oculopharyngeal muscular dystrophy through muscle atrophy' to PLOS Genetics.

The manuscript was fully evaluated at the editorial level and by independent peer reviewers. The reviewers appreciated the attention to an important topic but identified some concerns that we ask you address in a revised manuscript

We therefore ask you to modify the manuscript according to the review recommendations. Your revisions should address the specific points made by each reviewer.

[LINK]

Yours sincerely,

Bingwei Lu

Associate Editor

PLOS Genetics

Gregory Barsh

Editor-in-Chief

PLOS Genetics

Reviewer's Responses to Questions

**Comments to the Authors:**

Reviewer #2: The authors have responded very well to my initial suggestions, including producing new experimental data. Most notably, they include a new series of fluorescent images of indirect flight muscles, which are also quantified as to phenotype. Some of the English usage is not precise in the revisions, but presumably will be corrected by a copy editor. Below are a few minor points that should be attended to for clarity:

Page 11: Regarding the Rpn10 mutant, it is possible that the portion of the transcript that was tested by RT-PCR was produced at normal levels, but the P element in the coding sequence prevented normal translation. This may be what the authors meant by indicating it could be a null allele, but the explanation given in my first sentence would clarify that for the reader.

Page 11: Explain that the mutant lines reduced the proteins in all tissues, so that is why you did the muscle-specific knockdown.

Page 13 top: Change “checked” to “verified”. The word checked does not indicate what the outcome of the experiment was.

Figure 3E legend: exemplified not examplified (Figure 4C as well). Anti-Kettin appears magenta rather than red in the image I was provided.

Reviewer #3: Dear Authors,

The revised manuscript contains additional data supporting the contribution of increased ubiquitin-proteasome activity to muscle atrophy in a Drosophila model of Oculopharyngeal Muscular Dystrophy. The authors have rather satisfactorily answered my suggestions and the criticisms and suggestions by other reviewers. Thanks for this.

Among the revisions which significantly improved the ms:

- The added S1 figure convincingly shows the strong muscle defects in 3rd instar larvae, contrasting with the absence of muscle defects in late embryos, indicative of a progressive muscle atrophy.

- Fig.3 E and F (and 3C) nicely illustrate the progressivity of the disease.

- Fig.4, 5, 6. Additional data showing reduced proteasome activity in transheterozygotes help interpreting differences in rescue scores between Rpn10 and Rpn11/Prosb4/Pomp

- Fig.8. the inset: “yet compatible with PABPN17-Ala oligomer degradation except for rpn10” reflects well some seemingly contradictory data.

- Discussion. One added sentence (last of the second-to-last paragraph): “this sequence of events might participate in the progressivity of the disease as proteasome upregulation would be subsequent to defects arising earlier, namely oxidative stress and accumulation of PABPN1 oligomers that might slowly build up in patients” summarizes some remaining questions (see below).

Before publication, I would suggest a few more corrections.

- Introduction: extend “we use a Drosophila model of OPMD” to: “we use a Drosophila model of OPMD induced by mesodermal expression of PABPN1” to make clear that it is not a genetic model.

- Discussion: The authors might elaborate more on the added sentence: “this sequence of events might participate in the progressivity of the disease as proteasome upregulation would be subsequent to defects arising earlier, namely oxidative stress and accumulation of PABPN1 oligomers that might slowly build up in patients” using, for example, references abundantly quoted earlier in the discussion (23, 24, 65). As it is, extensive description of data from OPMD mouse models is more introduction than discussion of the new data reported here.

- New Fig.S1. The figure legend could be more precise and help readers by: i) indicating that 3 adjacent hemisegments are shown. ii) highlighting one specific muscle by its name in the larval panels, (for example LL1 which is well visible and missing in one segment in the right-most panel).

Reviewer #4: Interesting work.

**Have all data underlying the figures and results presented in the manuscript been provided?**

Reviewer #2: Yes

Reviewer #3: Yes

Reviewer #4: None

PLOS authors have the option to publish the peer review history of their article (what does this mean?). If published, this will include your full peer review and any attached files.

Reviewer #2: **Yes: **Sanford I. Bernstein

Reviewer #3: **Yes: **Alain VINCENT

Reviewer #4: No

---

## [Editor Report · Decision Letter 2]

1 Jan 2022

Dear Dr Simonelig,

We are pleased to inform you that your manuscript entitled "Activation of the ubiquitin-proteasome system contributes to oculopharyngeal muscular dystrophy through muscle atrophy" has been editorially accepted for publication in PLOS Genetics. Congratulations!

Yours sincerely,

Bingwei Lu

Associate Editor

PLOS Genetics

Gregory Barsh

Editor-in-Chief

PLOS Genetics

Comments from the reviewers (if applicable):

**Data Deposition**

http://datadryad.org/submit?journalID=pgenetics&manu=PGENETICS-D-21-00502R2

**Press Queries**

---

## [Editor Report · Acceptance letter]

7 Jan 2022

PGENETICS-D-21-00502R2 

Activation of the ubiquitin-proteasome system contributes to oculopharyngeal muscular dystrophy through muscle atrophy 

Dear Dr Simonelig, 

We are pleased to inform you that your manuscript entitled "Activation of the ubiquitin-proteasome system contributes to oculopharyngeal muscular dystrophy through muscle atrophy" has been formally accepted for publication in PLOS Genetics! Your manuscript is now with our production department and you will be notified of the publication date in due course.

With kind regards,

Livia Horvath

PLOS Genetics

On behalf of:
